# The quorum sensing regulator RhlR positively controls the expression of the type III secretion system in *Pseudomonas aeruginosa* PAO1

Luis Fernando Montelongo-Martínez[1], Miguel Díaz-Guerrero[1,2], Verónica Roxana Flores-Vega[3], Martín Paolo Soto-Aceves[4], Roberto Rosales-Reyes[5], Sara Elizabeth Quiroz-Morales[4], Bertha González-Pedrajo[2], Gloria Soberón-Chávez[4], Miguel Cocotl-Yañez[1]*

1 Facultad de Medicina, Departamento de Microbiología y Parasitología, Universidad Nacional Autónoma de México, Ciudad de México, México, 2 Departamento de Genética Molecular, Instituto de Fisiología Celular, Universidad Nacional Autónoma de México, Ciudad Universitaria, Ciudad de México, México, 3 Departamento de Biomedicina Molecular, Centro de Investigación y de Estudios Avanzados del IPN, Ciudad de México, México, 4 Departamento de Biología Molecular y Biotecnología, Instituto de Investigaciones Biomédicas, Universidad Nacional Autónoma de México, Ciudad de México, México, 5 Facultad de Medicina, Unidad de Medicina Experimental, Universidad Nacional Autónoma de México, Ciudad de México, México

* mcocotl@comunidad.unam.mx

## Abstract

*Pseudomonas aeruginosa* is an opportunist bacterium that causes acute and chronic infections. During acute infections, the type III secretion system (T3SS) plays a pivotal role in allowing the bacteria to translocate effectors such as ExoS, ExoT, and ExoY into host cells for colonization. Previous research on the involvement of quorum sensing systems Las and Rhl in controlling the T3SS gene expression produced ambiguous results. In this study, we determined the role of the Las and Rhl systems and the PqsE protein on T3SS expression. Our results show that in the wild-type PAO1 strain, the deletion of *lasR* or *pqsE* do not affect the secretion of ExoS. However, *rhlI* inactivation increases the expression of T3SS genes. In contrast to the *rhlI* deletion, *rhlR* inactivation decreases both T3SS genes expression and ExoS secreted protein levels, and this phenotype is restored when this mutant is complemented with the *exsA* gene, which codes for the master regulator of the T3SS. Additionally, cytotoxicity is affected in the *rhlR* mutant strain compared with its PAO1 parental strain. Overall, our results indicate that neither the Las system nor PqsE are involved in regulating the T3SS. Moreover, the Rhl system components have opposite effects, RhlI participates in negatively controlling the T3SS expression, while RhlR does it in a positive way, and this regulation is independent of C4 or PqsE. Finally, we show that *rhlR*, *rhlI*, or *pqsE* inactivation abolished pyocyanin production in T3SS-induction conditions. The ability of RhlR to act as a positive T3SS regulator in the absence of its cognate autoinducer and PqsE shows that it is a versatile regulator that controls different virulence traits allowing *P. aeruginosa* to compete for a niche.

**Data Availability Statement:** All relevant data are within the manuscript and its Supporting Information files.

**Funding:** MC-Y research was supported by Consejo Nacional de Humanidades, Ciencias y Tecnologías (CONAHCYT) FORDECYT- PRONACES grant 53366 and Programa de Apoyo a Proyectos de Investigación e Innovación Tecnológica (PAPIIT) DGAPA, Universidad Nacional Autónoma de México (UNAM), grant number IA204221 and IA200823. GS-Ch and BG-P research were supported by DGAPA, PAPIIT UNAM grant IN201222 and IN229023, respectively. The funders had no role in study design, data collection and analysis, decision to publish, or preparation of the manuscript.

**Competing interests:** The authors have declared that no competing interests exist.

## Introduction

*Pseudomonas aeruginosa* is a human opportunist pathogen that causes acute and chronic infections. This bacterium possesses a vast arsenal of virulence factors including pyocyanin, rhamnolipids, and elastase, that allows it to compete for a favorable niche [1,2]. One of the main virulence determinants in *P. aeruginosa* acute infection is the type III secretion system (T3SS), involved in avoiding phagocytosis and inducing macrophage apoptosis, among others roles [3–5]. Mutants defective in the T3SS machinery are less virulent in a mouse model infection [6,7].

The T3SS resembles a molecular syringe, known as injectisome, which forms a channel that crosses the bacterial envelope and the host cell membrane, enabling bacteria to inject effectors such as ExoS, ExoT and ExoY into the host cell cytoplasm [5,8,9]. ExoS and ExoT share 76% of identity in amino acid sequence and inhibit phagocytosis by disrupting actin cytoskeletal arrangement and signal transduction cascades essential for phagocytic function [10]. ExoY possesses adenylate cyclase activity that increases cAMP intracellular levels damaging the cellular function [11]. The genes encoding the structural and regulatory elements of the injectisome are organized together in five operons, whereas the genes that encode for the effectors and associated chaperones are distributed in the chromosome [12]. Expression of the T3SS genes is tightly regulated and is induced when *P. aeruginosa* is in contact with host cells or when extracellular calcium concentrations are reduced (S1 Fig) [13,14]. In addition, all T3SS genes are regulated at the transcriptional level by the main activator ExsA, a member of the AraC family of transcriptional regulators [15,16]. Each of the five operon promoters and the promoters of the effectors contain an ExsA binding motif located ~45 bp upstream of the transcriptional start site [17]. *exsA* is expressed within *exsCEBA* operon, whose transcription is positively controlled by PsrA and negatively by MvaT [18,19]. Also, *exsA* contains an internal promoter that is transcriptionally activated by Vfr and negatively regulated by MvaT and MvaU [20,21]. Furthermore, it was recently reported that spermidine modulates the T3SS by affecting *exsCEBA* operon expression [22,23]. An additional layer of regulation includes a partner-switching mechanism that involves the antiactivator ExsD and the antiantiactivator ExsC, which together control ExsA activity [24].

In *P. aeruginosa*, the synthesis of several virulence factors is regulated at the transcriptional level by the quorum sensing (QS) response, which is a cellular communication process based on the production and secretion of signal molecules named autoinducers (AI) that when are extracellularly accumulated, enter into the bacterium and bind to a transcriptional regulator that turns on virulence genes expression [25]. This bacterium harbors three QS systems named Las, Rhl, and Pqs that, in certain conditions such as rich medium, are hierarchically organized with the Las system placed at the top of this regulatory network [26–28]. The first two QS systems are based on the synthesis and detection of N-acyl-homoserine lactones while the Pqs system is based on producing and detecting alkyl-quinolones molecules. In the Las system, LasR is the transcriptional regulator that binds to N-3-oxo-dodecanoyl-homoserine lactone (C12), which is synthesized by LasI, to activate the expression of some virulence factors, and also of *rhlR* and *rhlI*, which encode to the transcriptional factor RhlR and the N-butyryl-homoserine lactone (C4) synthase, RhlI [29,30]. RhlR binds to C4 and activates the expression of genes involved in pyocyanin and rhamnolipids synthesis, among others [31,32]. In the Pqs system, PqsR is the transcriptional regulator that binds to 2-heptyl-3-hydroxy-1H-quinolin-4-one (PQS) or 2-heptyl-1H-quinolin-4-one (HHQ) and controls pyocyanin production by turning on the expression of *pqsE* [28,33]. In this regard, it has been reported that in addition to C4, RhlR can work by forming a complex with PqsE to control the expression of a different subset of genes; therefore, there are RhlR regulons dependent and independent of C4 [32,34–36].

Moreover, it has been reported that RhlR can control the expression of genes in the absence of its autoinducer [37]. Thus, these characteristics make RhlR a more versatile regulator than LasR.

Since the Las and Rhl QS systems control about 6% of gene expression [30], it has been investigated whether these two systems also control the T3SS gene expression. The first report established that *exoS* expression increased when *rhlI* or *rhlR* were inactivated, suggesting that the Rhl system negatively controls the T3SS; while *exoS* transcription was not affected when *lasI* was inactivated [38]. In a second study, it was reported that *rhlI* inactivation increased the expression of some T3SS genes and ExoS was secreted in the early stages of growth. In this study, the authors reported that *lasR* inactivation did not affect the expression of the T3SS genes indicating that the Rhl system, but not the Las system, is a negative regulator of the T3SS. Moreover, it was also reported that the *exsCEBA* operon, which encodes for ExsA activator, was not up-regulated in the *rhlI* mutant strain, suggesting that the Rhl system individually controls each promoter of the T3SS genes [39]. On the other hand, in recent reports, it was documented that even though a *lasR*/*rhlR* double mutant strain abolishes the production of elastase, pyocyanin, and rhamnolipids, the T3SS was not affected [40]. Furthermore, when the PAO1, PA14 and two clinical isolates were treated with the AiiM lactonase, which can degrade C12 and C4, the virulence factors production such as elastase, pyocyanin, and HCN was reduced but the secretion of exotoxins ExoS and ExoU was similar to that of the wild-type strain [41]. These data suggested that the T3SS expression is not regulated by the QS response. Additionally, it has been reported that *rhlI* mutants display higher virulence compared with *rhlR* mutants, and in the latter, the virulence is attenuated [32,42]. Therefore, there are some discrepancies in the virulence and regulation of the T3SS by the QS systems.

Herein, we determine the effect of the QS systems on the expression of the T3SS. We found that neither *lasR* nor *pqsE* inactivation affects the ExoS protein levels. However, regarding the Rhl system, our results showed that the inactivation of *rhlI* up-regulates the expression of T3SS genes, while *rhlR* inactivation down-regulates it. Moreover, the *rhlR* mutant strain showed reduced cytotoxicity, which is in line with previous reports where virulence is affected when *rhlR* is inactivated. Also, here we show that constitutive expression of the *PA2592* gene, which codes for a putative spermidine-binding protein, partially restores ExoS secretion in the *rhlR* mutant strain in the early stationary phase. Finally, we show that even though RhlR controls T3SS expression in the absence of C4 and PqsE, both are required for the positive control of RhlR on pyocyanin synthesis in T3SS-induction conditions.

## Materials and methods

### Bacterial strains and growth conditions

The bacterial strains and plasmids used in this study are listed in supplemental material S1 Table. *Pseudomonas aeruginosa* MPAO1 strain was used in all the experiments (referred to as PAO1) whereas *Escherichia coli* DH5a strain was used for standard techniques of cloning and propagation. Unless otherwise noted, *P. aeruginosa* and *E. coli* strains were grown in LB (Lysogenic- Broth) medium at 37°C and 225 rpm. For expression assays, LB medium was used as a non-induction medium, and LB medium supplemented with 5 mM EGTA and 20 mM MgCl$_2$ was used for T3SS induction [43]. When necessary, antibiotics at the following final concentrations were used for *P. aeruginosa*: tetracycline (Tc) 120 µg/mL, streptomycin (Sm) 200 µg/mL, carbenicillin (Cb) 200 µg/mL, apramycin (Apc) 150 µg/mL, gentamicin (Gm) 100 µg/mL. For *E. coli*: tetracycline (Tc) 15 µg/mL, streptomycin (Sm) 30 µg/mL, ampicillin (Amp) 200 µg/mL and gentamicin (Gm) 15 µg/mL.

## DNA manipulation techniques

The genomic DNA template was obtained from strain PAO1 using the GeneJET DNA purification system (Thermo Scientific). The high-fidelity DNA polymerase enzyme Phusion (Thermo Scientific) was used to amplify the DNA regions. DNA fragments were obtained from agarose gel bands and DNA was purified using the Wizard SV Gel and PCR Clean-Up System protocol (Promega). Restriction enzymes (New England Biolabs) and T4 DNA ligase enzyme (Promega) were used according to manufacturer instructions. Plasmids were purified using Wizard Plus SV Minipreps DNA Purification Systems (Promega) and manipulated according to standardized techniques [44]. Synthesis of oligonucleotide and DNA sequencing were performed at Unidad de Síntesis y Secuenciación de DNA (USSDNA) by Instituto de Biotecnología at Universidad Nacional Autónoma de México (UNAM). Oligonucleotide pairs used for the PCR reactions are listed in S2 Table.

## Construction of transcriptional fusions

Mini-CTX-lux plasmid [45] was used to construct all transcriptional fusions. The promoter sequences were amplified at 60°C using PAO1 genomic DNA and specific oligonucleotides pair (S2 Table) that include the -35/-10 binding sites for RNA polymerase and the ExsA-binding motif (S2 Fig). PCR products were purified and cloned into the XhoI and HindIII sites in mini-CTX-lux. Also, the pCTX plasmid was used as a negative control for *lux* expression, which was constructed by cloning ~400 bp of the *rhlR* structural region, previously amplified with primers pair rt-rhlR-F3 and rt-rhlR-R2, into the SmaI site from mini-CTX-lux. Plasmids were mobilized into *P. aeruginosa* strains and chromosomal integration were confirmed by PCR reactions, using a forward primer, corresponding to the region of each cloned promoter, and a LuxRv reverse primer recognizing the *luxC* gene of mini-CTX-lux plasmid.

## Construction of pExsA and pUC2592 plasmids

A 975 bp and 1,232 bp corresponding to the structural region of the transcriptional regulator *exsA* and the gene *PA2592* were amplified using the primers pair ExsA_Fw/ExsA_Rv and 5UpEcIPA2592Fw/3DwBmIPA2592Rv, respectively (S2 Table). The corresponding products were purified and cloned into the BamHI-HindIII and EcoRI-BamHI sites in pUCP20 plasmid [46], obtaining pExsA and pUC2592 plasmids. Then, one μg of the purified pExsA or pUC2592 plasmid was introduced by electroporation into the PAOΔ*rhlR* strain.

## Generation of mutant strains

Gene deletions were performed by homologous recombination to the bacterial chromosome of plasmid-borne insertion-deletion as previously described [47] with minor modifications. Briefly, allele replacement of the *lasR* gene from PAO1 was achieved by constructing the pEX-lasR::Apc deletion plasmid as follows: a 487 bp fragment and a 644 bp fragment corresponding to the upstream and downstream region of the *lasR* gene were amplified from PAO1 genomic DNA with primers pair 6709–2015_H3lasRUp/8653-2015_lasR5Apra and 6514–2018_lasR3Apra/6710-2015_H3lasRDown, respectively. Also, an apramycin resistant cassette was amplified from the pIJ773 plasmid [48] with primers 9522–2014_F-Apra and 9523–2014_R-Apra. The three PCR products were purified and used as a template in a nested PCR. The PCR product was digested with HindIII and cloned into the HindIII site of the pEX18 plasmid [49], which is unable to replicate in *P. aeruginosa*, resulting in the pEX-lasR::Apc plasmid. This plasmid was mobilized into the wild-type PAO1 strain to replace the *lasR* gene with the apramycin resistance marker by double allelic exchange, thus obtaining the *lasR* single mutant PAOΔ*lasR*

strain. Also, the pEX-lasR::Apc plasmid was mobilized into the PAOΔ*rhlR* strain [46] to obtain the double mutant strain PAOΔ*lasR*Δ*rhlR*. For *rhlI* deletion, the pEX-rhlI::Aa plasmid [46] was used to construct the PAOΔ*rhlI*^*Apc*, and the resistance marker was subsequently removed using the pFLP2 plasmid, as previously reported [49], obtaining the PAOΔ*rhlI* strain. To obtain the double mutant strain PAOΔ*rhlI*Δ*pqsE*, the pEX-pqsE::Gm plasmid [36] was mobilized into the PAOΔ*rhlI* strain. Finally, the genes *exsA*, *exsD*, *pscB* and part of *pscC* were deleted in PAO1 strain using the plasmid pJET1.2_T3SS::Apc. This plasmid was constructed as follows: a 608 bp fragment and a 773 bp fragment corresponding to the upstream region of the *exsA* gene and downstream region of the *pscB* gene were amplified from PAO1 genomic DNA with primers pair 6767–2023_T3SSUp/6768-2023_T3SS3aaApra and 6769–2023_T3SS5aaApra/6770-2023_T3SSDown, respectively. Also, an apramycin resistant cassette was amplified as previously described. The three PCR products were purified and used as a template in a nested PCR and cloned into the pJET1.2 plasmid (Thermo-Fischer), resulting in the pJET1.2_T3SS::Apc plasmid, which was then mobilized into the PAO1 strain to obtain the double recombinant PAOΔT3SS ^*Apc*^ strain. The resistance marker was subsequently removed using the pFLP2 plasmid obtaining PAO1ΔT3SS strain. In each case, the candidate clones were positive selected on LB medium with the respective antibiotics and confirmed by PCR and sequencing of the modified region.

## Luminescence assays

Bacterial overnight pre-cultures were diluted at an O.D.$_{600}$ = 0.05 in 125 mL flasks with 15 mL of non-induction and/or induction medium, which were incubated until reaching the final cell density corresponding to the log phase (O.D.$_{600}$ = 0.8) or early stationary phase (O.D.$_{600}$ = 2.0). For each biological assay, 200 μL samples of each culture were collected at the desired cell density and loaded in triplicate into 96-well flat clear bottom black polystyrene plates (Costar). The ratio of relative units of luminescence (R.L.U.) produced was quantified using the Synergy HT Plate Reader (Biotek) and normalized over the O.D.$_{600}$ value at the time of sample collection (R.L.U./O.D.$_{600}$). The results represent the mean ±S.D. of three biological experiments.

## Western blot assays

Bacterial overnight pre-cultures were diluted at an O.D.$_{600}$ = 0.05 in 125 mL flasks with 15 mL of induction or non-induction medium, which were incubated at 37˚C and 225 rpm, until the desired cell density was reached. Subsequently, 1 mL samples were collected and centrifuged at 4˚C (14,000 rpm, 2 min), and proteins of the supernatant were precipitated with 100 μL of trichloroacetic acid (TCA) 100% at 4˚C, overnight. Then, samples were centrifugated at 4˚C (14,000 rpm, 30 min), and pellets were resuspended with the volume corresponding to 30 μL of SDS-PAGE loading buffer previously normalized to the cell density value at the time of sample collection. Resuspended samples were denatured at 90˚C for 5 minutes. For electrophoresis separation, 5 μL of each sample was loaded in 12% SDS-PAGE gels following Bio-Rad protocols. Proteins were transferred to 0.2 μm nitrocellulose membranes and blocked with 5% BSA in Tris-HCl buffer pH 7.4 supplemented with 0.1% Tween-20 detergent (TBS-T) at 4˚C, for 2 h. The membranes were washed for 15 min with 15 mL of TBS-T, three times. Subsequently, the first antibody anti-ExoS [40] diluted 1:10,000 was added and incubated at 4˚C for 1 h. The membranes were washed, and then a second antibody was added: 1:10,000 anti-IgG-GAR (Jackson Immunoresearch) or 1:5000 anti-rabbit IgG conjugated to alkaline phosphatase (Abcam) and incubated at 4˚C for 1 h. Finally, the membranes with anti-IgG-GAR were washed and a reaction 1:1 solution of HRP chemiluminescent Immobilon Western kit (Millipore) was added and the bands were developed on X-ray film (Carestream MXB-Blue film).

Membranes with anti-rabbit IgG conjugated to alkaline phosphatase (anti-IgG-AP) were washed and developed using a 1-step NBT/BCIP solution (Thermo Scientific). GroEL was used as a loading control since it is constitutively expressed in the conditions used [50]. For GroEL detection the same membranes were washed with 15 mL of stripping buffer (25 mM glycine, 1% SDS, pH 2.0) at 4˚C for 1 h with shaking and washing. The membranes were blocked and incubated for 12 h at 4˚C, washed, and incubated with 1:10,000 anti-GroEL poly-clonal antibody (Sigma). Finally, the membranes were washed with TBS-T and incubated with 1:10,000 anti-IgG-GAR or 1:5,000 anti-IgG-AP. Bands were revealed on an X-ray film or using a 1-step NBT/BCIP solution.

## Cytotoxicity assays

Strains of *P. aeruginosa* were grown on in shaking, 180 rpm at 37˚C in LB broth. The bacteria were adjusted at an MOI of 600 nm to infect $6\times10^5$ HeLa cells (ATCC® CCL-2TM). To syn-chronize the infection, the plates were centrifuged at 1,400 rpm for 2 min at room temperature, and incubated at 37˚C, 5% $CO_2$, and 5% humidity for 30 min. Next, the infected cells were washed and incubated for 24 hours at 37˚C, 5% $CO_2$, and 5% humidity. The supernatants of infected cells were collected and centrifuged at 14,000 rpm for 2 minutes. The clarified super-natants were used to quantify the cytosolic enzymatic activity of lactate dehydrogenase (Pro-mega). For this, 50 μL of the supernatants were taken, and mixed with 50 μL of the substrate in 96-well flat-bottomed plates (Nunc). The mixture was incubated for 10–20 minutes at room temperature, and the enzymatic activity was quantified spectrophotometrically at 490 nm. Cytotoxicity was quantified using the following formula: % CTX = [(sample cytotoxicity-spon-taneous cytotoxicity) / (total cytotoxicity-spontaneous cytotoxicity)] x 100% [51].

## Pyocyanin quantification

Bacterial overnight pre-cultures were diluted at an O.D.$_{600}$ = 0.05 and incubated in 125 mL flasks with 15 ml of induction medium for 24 h with continuous shaking (225 rpm) at 37˚C. Pyocyanin production was quantified from cell-free supernatants at 695 nm and normalized by the O.D.$_{600}$ at 24 h, as reported previously [32].

## Densitometry analysis

Densitometry analysis of at least three Western blot assays was carried out using the ImageJ software [52]. Relative ExoS protein levels were normalized by the levels of GroEL, used as a loading control. ExoS protein levels of PAO1 strain were considered as 100% and ATCC 9027 as 0%.

## Statistical analysis

Means and standard deviation from at least three biological replicates were analyzed using the Graph Prism 9.0 statistical software with a confidential level of 95% (α = 0.05%). Data were considered statistically significant if the difference value was p<0.05.

## Results

### RhlR, but not LasR, decreases ExoS secretion

In order to determine whether the QS regulators, LasR and RhlR, are able to control the expression of the T3SS, we measured ExoS secretion by Western blot using PAOΔ*lasR* and PAOΔ*rhlR*, and a double mutant PAOΔ*lasR*Δ*rhlR* strain, and compared it with the wild-type PAO1 strain. In addition, ATCC 9027 strain was used as a negative control since this strain

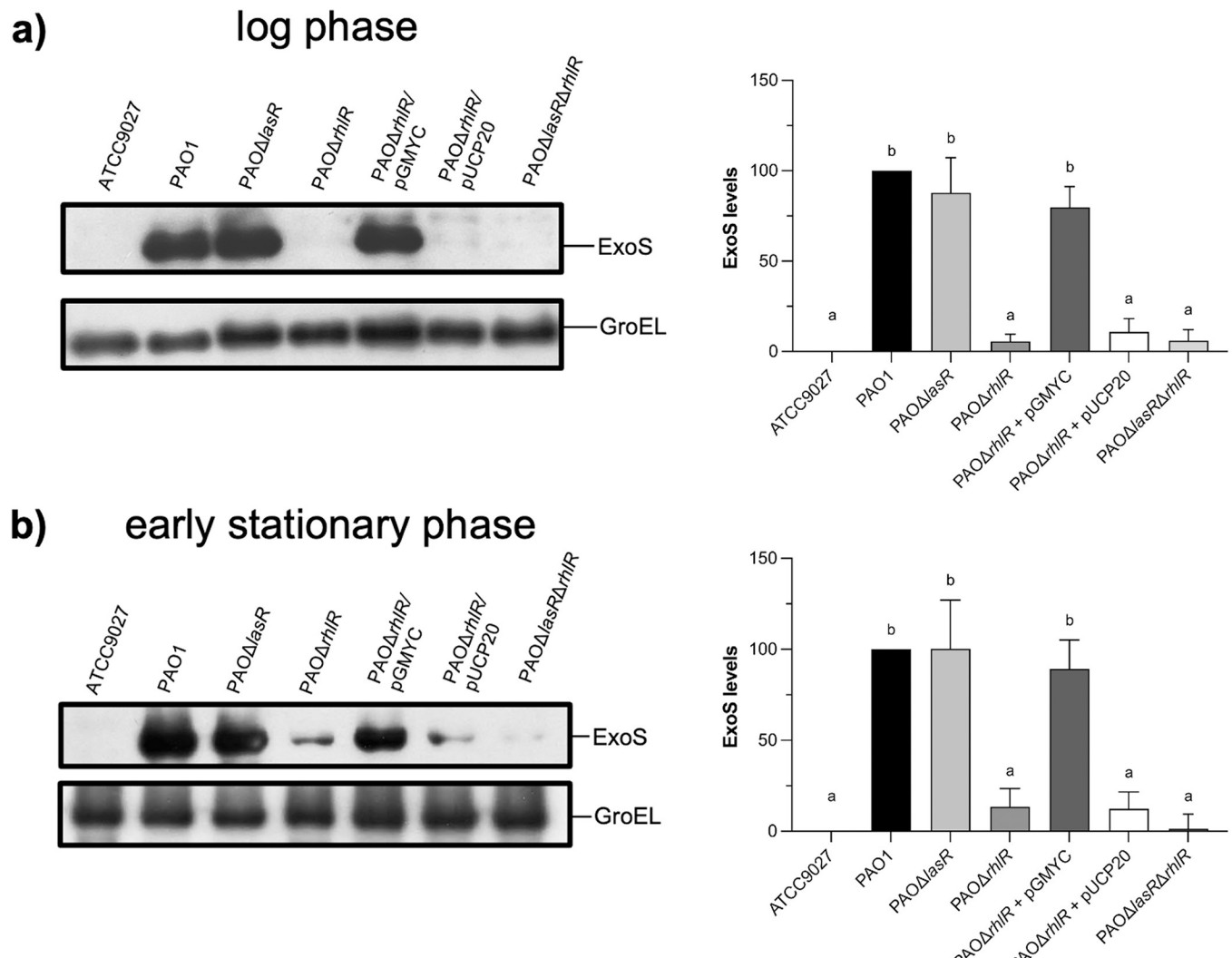

**Fig 1. Effect of *lasR* and *rhlR* inactivation on ExoS secretion.** ExoS identification was performed by Western blot assay using anti-ExoS polyclonal antibody on supernatants of strains grown in induction conditions at log phase (a) and early stationary phase (b). PAO1 was used as a positive control, whereas ATCC 9027, lacking the T3SS, was used as a negative control. GroEL was detected using polyclonal antibody anti-GroEL and used as a loading control. The densitometry graphs show the mean ± S.D. of ExoS levels of at least three biological replicates. Significant differences were obtained by ordinary one-way ANOVA and Tukey's multiple comparisons ($\alpha = 0.05\%$). Different letters indicate significant differences, while equal letters indicate no significant differences.

has a natural deletion of the T3SS [40]. Western blot assays were carried out in induction conditions at an O.D.$_{600}$ of 0.8 and 2.0 which correspond to the log phase and early stationary phase, respectively. As shown in Fig 1A, in the log phase, ExoS protein secretion levels were similar in the PAOΔ*lasR* strain compared to the wild-type PAO1 strain, while in the PAOΔ*rhlR* strain ExoS is barely detected. In addition, and similar to *rhlR* inactivation, ExoS secreted protein levels were almost abolished in the double mutant PAOΔ*lasR*Δ*rhlR* strain. These results were similar to those of the early stationary phase where ExoS protein secretion levels were not affected by *lasR* deletion but decrease in both the *rhlR* mutant strain and the double *lasR*/*rhlR* mutant strain (Fig 1B). Thus, since RhlR seems to have a positive effect on *exoS* expression, we verified that this phenotype was complemented with *rhlR* using pGMYC plasmid carrying this gene. As shown in Fig 1, the *rhlR* mutant strain with pGMYC plasmid fully restored the ExoS secretion, but not with the empty plasmid pUCP20 used as a negative

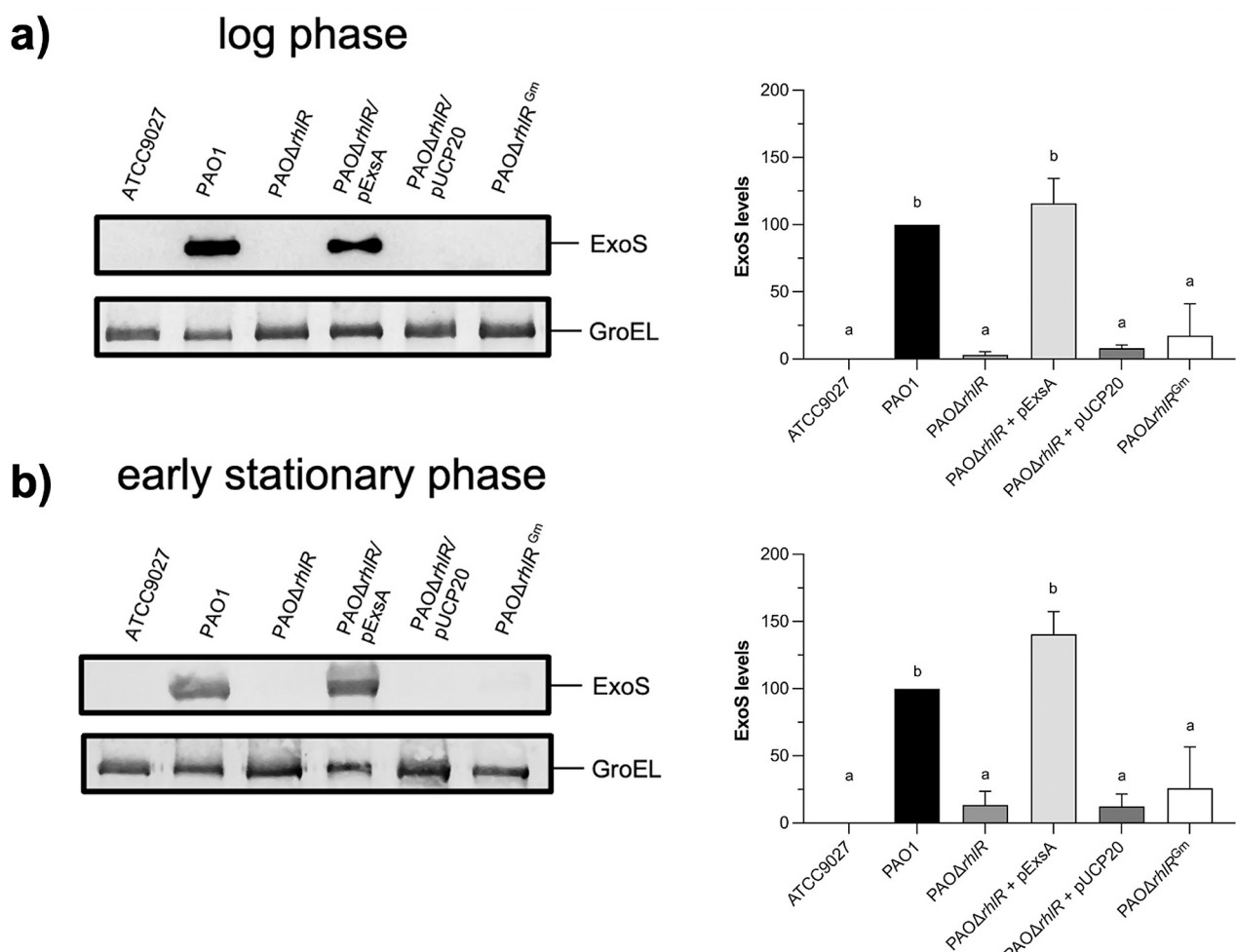

**Fig 2. Effect of *exsA* expression in the *rhlR* mutant strain on ExoS secretion.** ExoS identification was performed by Western blot assay using anti-ExoS polyclonal antibody on supernatants of strains grown in induction conditions at log phase (a) and early stationary phase (b). PAO1 was used as a positive control, whereas ATCC 9027, lacking the T3SS, was used as a negative control; also, PAOΔ*rhlR*$^{Gm}$ strain was included. GroEL was detected using polyclonal antibody anti-GroEL and used as a loading control. The densitometry graphs show the mean ± S.D. of ExoS levels of at least three biological replicates. Significant differences were obtained by ordinary one-way ANOVA and Tukey's multiple comparisons analysis (α = 0.05%). Different letters indicate significant differences, while equal letters indicate no significant differences.

control. With regard to the double mutant strain, *rhlR* expression restored the phenotype to that of the *lasR* mutant strain where ExoS is secreted (S3 Fig). Since these results contrast with the previous report where RhlR seemed to act as a negative regulator of *exoS* expression [38], we used another *rhlR* mutant strain with a different marker, PAO1Δ*rhlR*$^{Gm}$ [53], and Western blot was carried out to detect ExoS protein, producing similar results as the assays using the initial strain PAO1Δ*rhlR* (Fig 2). Together, these results suggest that RhlR positively regulates the expression of the T3SS.

## RhlR positively regulates the expression of *exoS*, *spcS* and the *exsCEBA* operon

Since *rhlR* inactivation almost abolishes ExoS secretion, we constructed an *exoS* transcriptional fusion (PexoS::lux) fusing the *exoS* promoter to the *luxCDABE* operon. Also, three additional transcriptional fusions were constructed using the promoter of the *spcS* gene which encodes

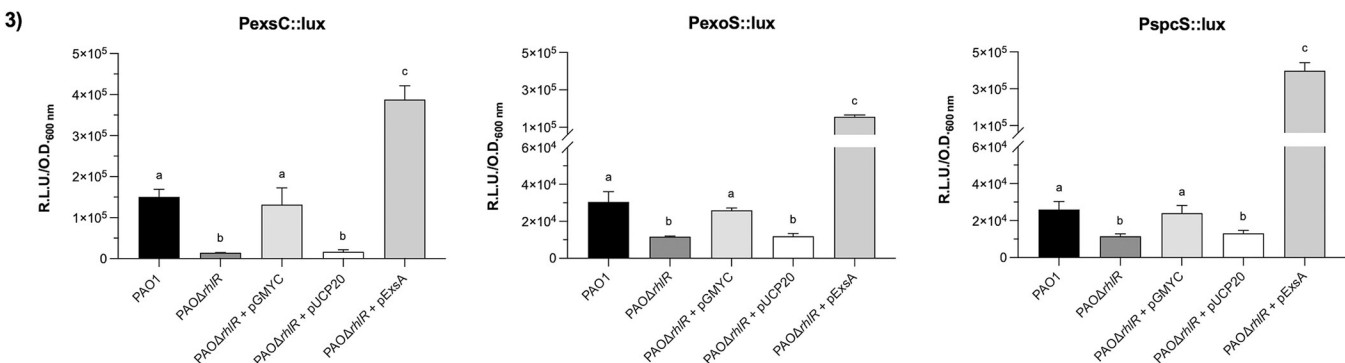

**Fig 3. Effect of *rhlR* inactivation on transcription of T3SS genes and *exsA* expression in the *rhlR* mutant strain.** Expression of the *exsCEBA* operon (PexsC::lux), *exoS* (PexoS::lux) and *spcS* (PspcS::lux) genes were evaluated by lux-transcriptional fusions. Strains were incubated in 15 ml of induction medium at 37°C and 225 rpm. Relative luminescence units (R. L.U.) were quantified and normalized to the O.D.$_{600}$ at the time of cell collection. Results represent the mean ± S. D. of three biological experiments performed in three replicates each time. Significant differences were obtained by ordinary one-way ANOVA and Tukey's multiple comparisons analysis (α = 0.05%). Different letters indicate significant differences, while equal letters indicate no significant differences.

for the ExoS chaperone (PspcS::lux), the promoter of the first gene of the *exsCEBA* operon (PexsC::lux), and the internal promoter of *exsA* (PexsA::lux). First, we tested whether these transcriptional fusions were activated in the T3SS-induction conditions as described in the methods section. As shown in S4 Fig, all fusions, except PexsA::lux, were activated when Ca$^{+}$ concentrations were reduced with EGTA during induction conditions. Then, PexoS::lux, PspcS::lux, and PexsC::lux plasmids were mobilized into the *rhlR* mutant strain and its derivatives, and luminescence was measured at an O.D.$_{600}$ of 0.8. As shown in Fig 3, the luminescence of the three transcriptional fusions was reduced in the *rhlR* mutant background compared to the wild-type strain, and the activity was restored when PAOΔ*rhlR* strain was complemented with pGMYC but not with the empty plasmid. The above results suggest that RhlR activates the expression of the T3SS genes.

## *exsA* expression restores ExoS secretion and the T3SS genes expression in the *rhlR* mutant strain

Since ExsA is the master regulator of the T3SS, we explored whether RhlR controls the T3SS expression via ExsA activity or whether it is an ExsA-independent regulation. In the first scenario, *exsA* expressed under a constitutive promoter must restore *exoS* transcription and also ExoS secretion in the *rhlR* mutant strain, but if this was not the case, then RhlR must control each operon or gene of the T3SS. Thus, we constructed a plasmid expressing *exsA* under a constitutive promoter, named pExsA, and it was mobilized into the PAOΔ*rhlR* strain. Then, we carried out a Western blot assay to detect ExoS protein in this strain. As shown in Fig 2, *exsA* constitutive expression in the *rhlR* mutant strain resulted in the detection of ExoS protein at an O.D.$_{600}$ of 0.8 and 2.0. Next, we measured the transcriptional activity of *exoS*, *exsCEBA*, and *spcS* in this background and compared it to the wild-type strain. As shown in Fig 3 the expression of the three transcriptional fusions was restored in the *rhlR* mutant strain when *exsA* is expressed and these levels are higher than in the wild-type strain. Since these results did not discard the possibility that RhlR and ExsA could act independently, we determined whether *rhlR* overexpression in a strain defective in the T3SS, named PAOΔT3SS, was able to restore *exsCEBA* and *exoS* expression in the log and early stationary phase. As shown in S5 Fig, the transcriptional activity of both fusions, PexsC::lux and PexoS::lux, in the PAOΔT3SS strain overexpressing *rhlR* was similar to those of the mutant strain and the mutant strain with the empty plasmid pUCP20, whereas in the wild-type strain, both transcriptional fusions were

activated. These results show that RhlR is unable to activate the T3SS expression in the absence of the master regulator ExsA and indicate that RhlR controls the T3SS by controlling *exsA* expression from the *exsCEBA* operon.

## Cytotoxicity is affected when *rhlR* is inactivated

PAO1 displays lower cytotoxicity compared to the PA14 strain due to its genome containing the *exoS* gene instead of *exoU*; however, ExoS is able to cause cell rounding and apoptosis in eukaryotic cells [5,54]. Thus, we carried out a cytotoxicity assay to determine whether *rhlR* inactivation, which reduces T3SS gene expression, also reduces cytotoxicity. Cell cultures from PAO1, PAOΔ*rhlR*, and its derivates were used to infect the HeLa eukaryotic cell line, and cytotoxicity was measured as described in the methods. As shown in Fig 4, in the *rhlR* mutant strain and complemented with the empty plasmid cytotoxicity was reduced compared to PAO1 strain or PAOΔ*rhlR* mutant strain complemented with *rhlR*. Moreover, *exsA* expression increased cytotoxicity in the *rhlR* mutant strain. These results are in agreement with our initial observations indicating that RhlR is a positive regulator of the T3SS and that this regulation is achieved by controlling *exsA* expression.

## RhlR regulates T3SS expression in the absence of C4 or PqsE

Since RhlR activity is usually dependent on binding to C4 and on the activity of PqsE [32], we carried out Western blot analysis at an $O.D._{600}$ of 0.8 and 2.0 to detect ExoS when *rhlI*, *pqsE*, or both genes are inactivated. As shown in Fig 5, ExoS secretion in the three mutant strains was similar to that in the wild-type strain at an $O.D._{600}$ of 0.8 and 2.0. We next measured the activity of the *exsCEBA* promoter in these three mutant strains at an $O.D._{600}$ of 0.8 using the PexsC::lux plasmid. As expected, the activity of *exsCEBA* promoter was similar in the three mutant strains compared to the wild-type PAO1 strain (S6 Fig). Since it has been reported that *rhlI* inactivation increases the expression of *exoS* and *exoT* at an $O.D._{600}$ >1 [38,39], we constructed an *exoT* transcriptional fusion (PexoT::lux). Then, PexoS::lux and PexoT::lux were mobilized into the *rhlI* mutant strain, and promoter activity of these two transcriptional fusions and the *exsCEBA* transcriptional fusion (PexsC::lux) were measured at an $O.D._{600}$ of 2.0. As shown in Fig 6, the expression of the three transcriptional fusions was higher in PAOΔ*rhlI* than in the wild-type strain indicating that, as previously reported, RhlI seems to negatively regulate the expression of the T3SS. These results suggest that RhlR without C4 is a positive *exsCEBA* regulator, while RhlR coupled with C4 acts as a negative regulator of its transcription.

## *PA2592* expression partially restores ExoS secretion in the *rhlR* mutant strain

Our data indicates that RhlR positively regulates T3SS, probably by regulating *exsA* expression; however, we could not detect a *las-rhl* box in the *exsCEBA* promoter indicating that the regulation must be indirect. Previously it was reported that spermidine works as a signal that modulates T3SS expression [22,23]. Interestingly, *PA2592* which codes for a putative spermidine-binding protein was downregulated when *rhlR* was inactivated, and the promoter region of this gene contains a *las-rhl* box that could be recognized by RhlR [55]. Therefore, we determined whether the expression of *PA2592* from a constitutive promoter was able to restore ExoS secretion in the *rhlR* mutant strain. We constructed the plasmid pUC2592, as described in the methods section, and it was mobilized into PAOΔ*rhlR* strain. Then, Western blot assay against ExoS was carried out at an $O.D._{600}$ of 0.8 and 2.0. Fig 7 shows that ExoS secretion was partially restored only in the early stationary phase. This result suggests that RhlR regulation of *PA2592* is involved in T3SS expression but that an additional pathway is also involved.

**4)**

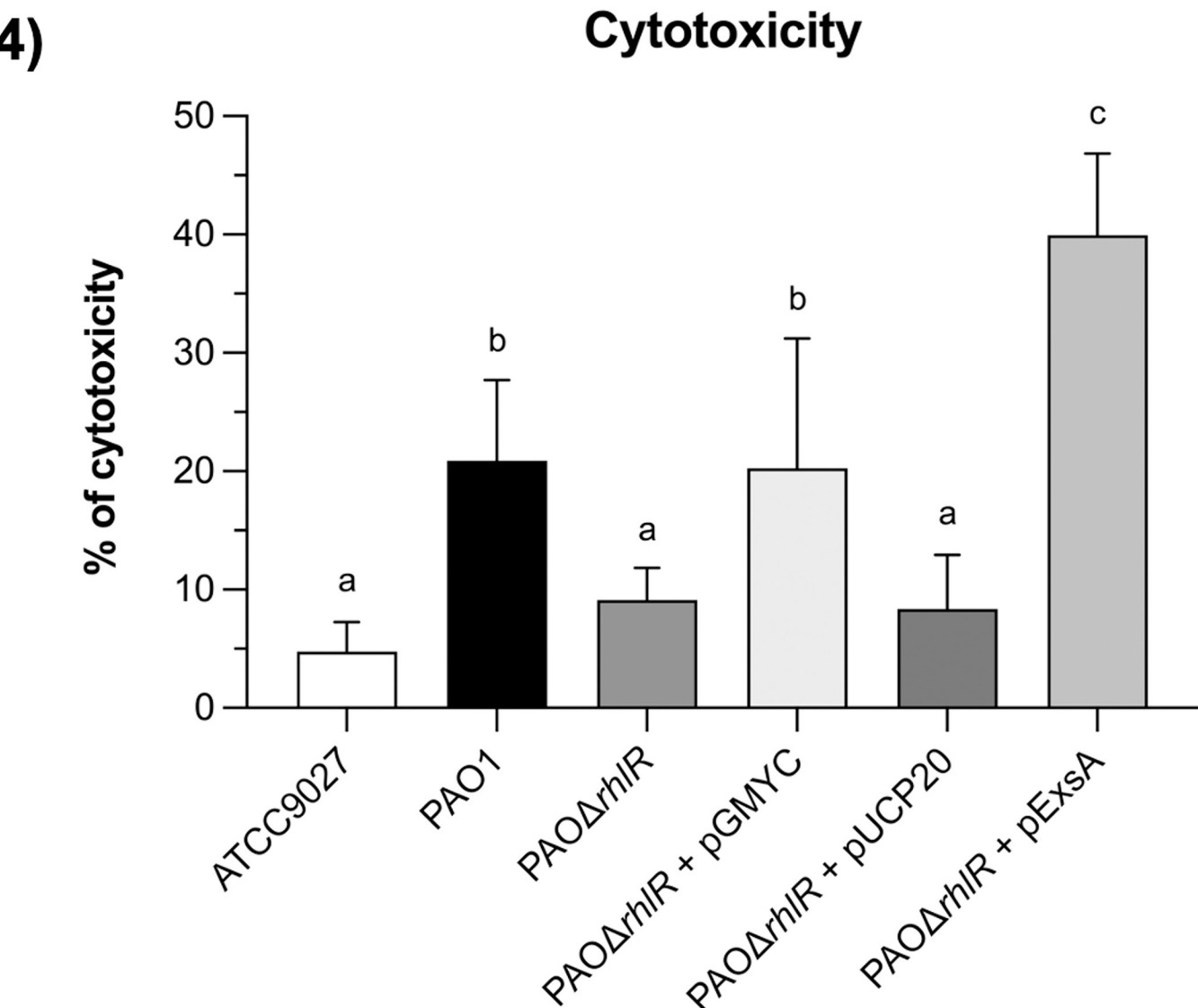

**Fig 4. Effect of *rhlR* inactivation on cytotoxicity.** Selected strains at MOI of 100 were used to infect 6x10⁵ HeLa cells (ATCC® CCL-2TM) in 24-well boxes. Infection conditions were synchronized and incubated at 37˚C, 5% CO2 and 5% humidity for 24 hours. Clarified supernatants were used to quantify the enzymatic activity of lactate dehydrogenase in 96-well flat-bottomed plates at 490 nm. Results represent the mean ± S.D. of three biological experiments performed in duplicates each time. Significant differences were obtained by ordinary one-way ANOVA and Tukey's multiple comparisons analysis (α = 0.05%). Different letters indicate significant differences, while equal letters indicate no significant differences.

### Pyocyanin production is dependent on C4 and PqsE in T3SS-induction conditions

As shown above, T3SS expression is regulated by RhlR in the absence of C4 or PqsE. Thus, we explored whether pyocyanin, whose synthesis is positively controlled by QS systems and particularly by RhlR, requires C4 and PqsE in T3SS-inducing conditions as required in LB medium [40,56]. The strains, including the *rhlR* mutant strain overexpressing *exsA*, were grown in T3SS-inducing conditions, and pyocyanin production was measured. As shown in Fig 8, *lasR* inactivation slightly reduced pyocyanin synthesis compared with the PAO1 strain. As expected, *rhlR* inactivation abolished pyocyanin production and it was restored in the PAOΔ*rhlR* strain when *rhlR* was expressed from pGMYC plasmid but not with the plasmid

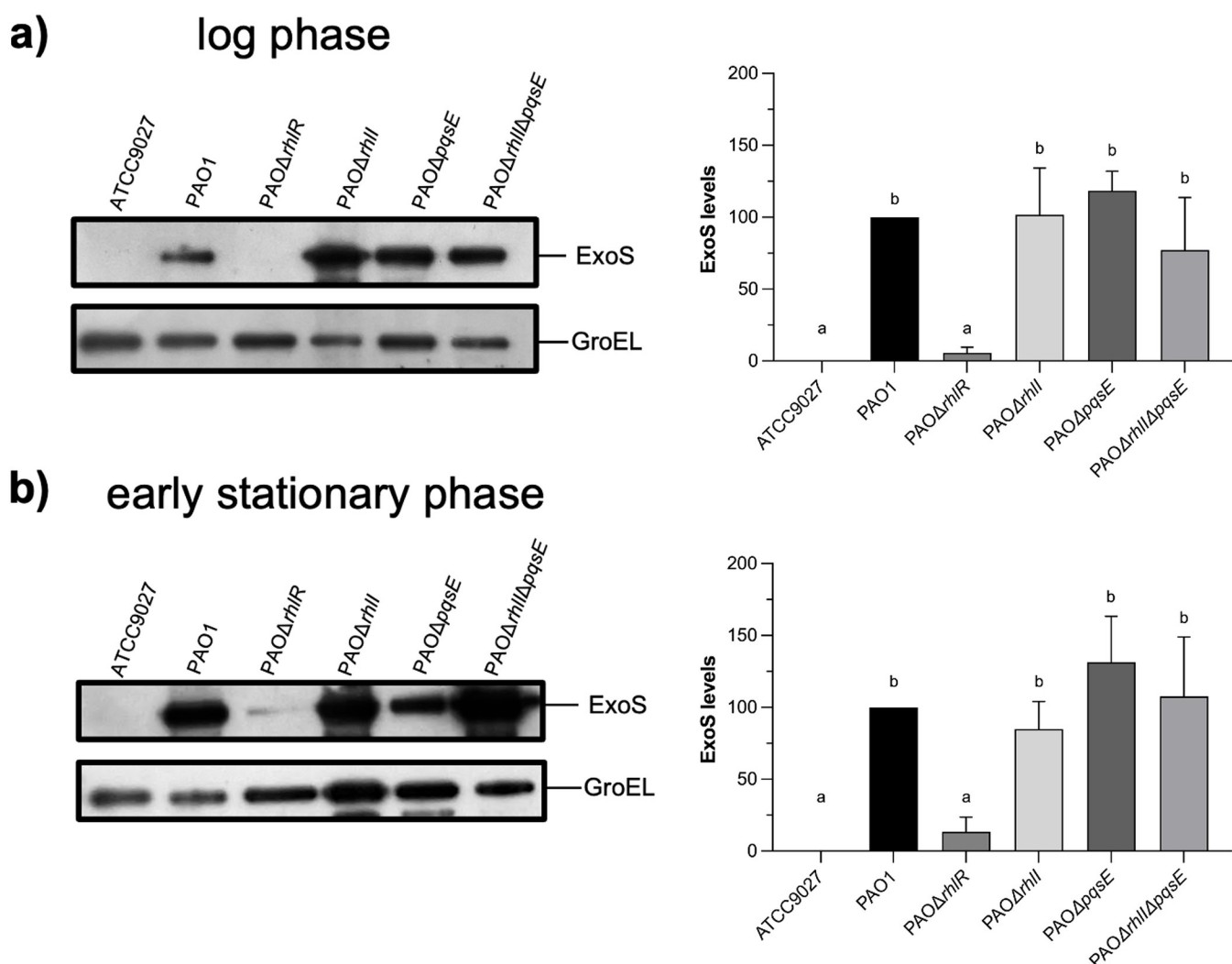

**Fig 5. Effect of *rhlI* and *pqsE* inactivation on ExoS secretion.** ExoS identification was performed by Western blot assays using an anti-ExoS polyclonal antibody on supernatants of strains grown in induction conditions at log phase (a) and early stationary phase (b). PAO1 was used as a positive control whereas ATCC 9027, lacking the T3SS, was used as a negative control. GroEL was detected using polyclonal antibody anti-GroEL and used as a loading control. The densitometry graphs show the mean ± S.D. of ExoS levels of at least three biological replicates. Significant differences were obtained by ordinary one-way ANOVA and Tukey's multiple comparisons analysis ($\alpha = 0.05\%$). Different letters indicate significant differences, while equal letters indicate no significant differences.

expressing *exsA* nor pUCP20 empty plasmid. Furthermore, *rhlI* and *pqsE* inactivation also abolishes pyocyanin synthesis since C4 and the PqsE protein have been reported to be necessary for the RhlR activity on pyocyanin production [56]. These results are similar to those previously reported in LB medium indicating that T3SS-inducing conditions do not affect pyocyanin regulation by the QS systems.

## Discussion

One of the main virulence determinants in *P. aeruginosa* is the T3SS which allows translocating effector proteins into the host cell to avoid phagocytosis [3,24]. PAO1 strains code for three exotoxins ExoY, ExoT, and ExoS while PA14 strain contains ExoU instead of ExoS, which makes it more cytotoxic [57,58]. The master regulator of this system is ExsA whose

**6)**

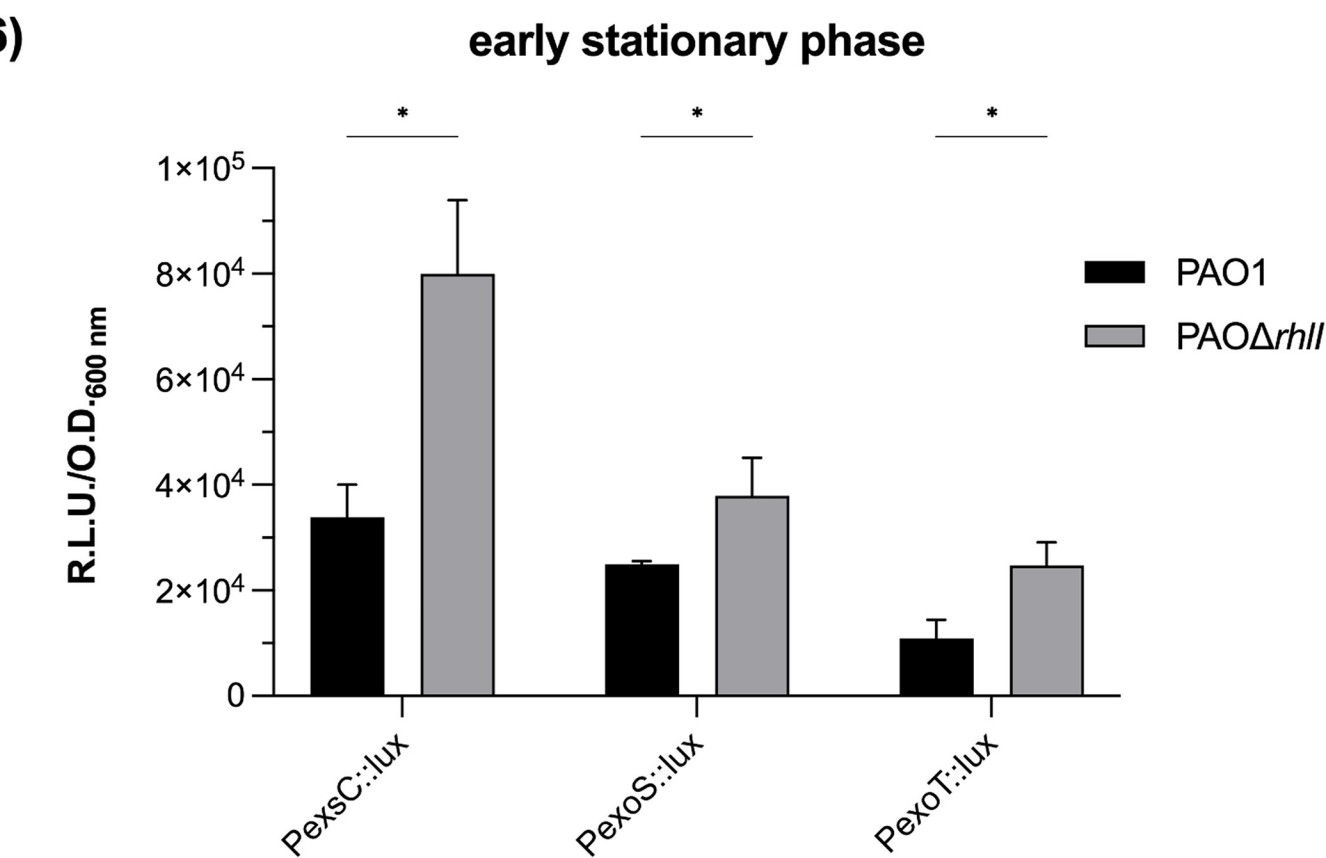

**Fig 6. Effect of *rhlI* inactivation on transcription of T3SS genes during early stationary phase.** Expression of the *exsCEBA* operon (PexsC::lux), *exoS* (PexoS::lux) and *exoT* (PexoT::lux) genes was evaluated by lux-transcriptional fusions. Strains were incubated in 15 ml of induction medium at 37°C and 225 rpm until reaching an O.D.$_{600}$ of 2.0. Relative luminescence units (R. L.U.) were quantified and normalized to the O.D.$_{600}$ at the time of cell collection. Results represent the mean ± S.D. of three biological experiments performed in three replicates each time. Significant differences were obtained by multiple unpaired *t*-test and are indicated with asterisks (* $p < 0.05$; ** $p < 0.01$; *** $p < 0.001$).

activation is coupled to a cascade of three interacting proteins (ExsC, ExsD, and ExsE) (S1 Fig) [24]. In addition, *exsA* expression is regulated at the transcriptional and post-transcriptional levels. The latter includes the Gac-Rsm system where RsmA is a post-transcriptional regulator that positively controls the translation of *exsA* [59]. Furthermore, transcription of *exsA* involves the activation of the *exsCEBA* operon by PsrA and it is negatively controlled by MvaT [18,19], while the internal *exsA* promoter is activated by Vfr and VqsM, and negatively regulated by MvaT and MvaU [20,21,60]. In addition to the gene regulation of the T3SS, other factors such as spermidine concentrations affect *exsCEBA* operon expression and, therefore, the T3SS activation [22,23].

Since QS systems are involved in regulating several virulence traits such as pyocyanin, elastase, and rhamnolipids, among others, different research groups have tried to elucidate whether QS systems are involved in controlling T3SS expression as well. Early reports placed the Rhl system, and particularly RhlI, as a negative regulator for the T3SS [38,39]. However, in recent years it has been reported that the QS systems seem not to be involved in controlling this secretion system [40,41]. Thus, our objective was to define the role of the QS system elements in T3SS. First, we verified that the T3SS is not induced in LB medium and is activated when EGTA is added during induction conditions (S4 and S7 Figs). Therefore, all experiments were carried out under T3SS-inducing conditions. Our results showed that LasR is not

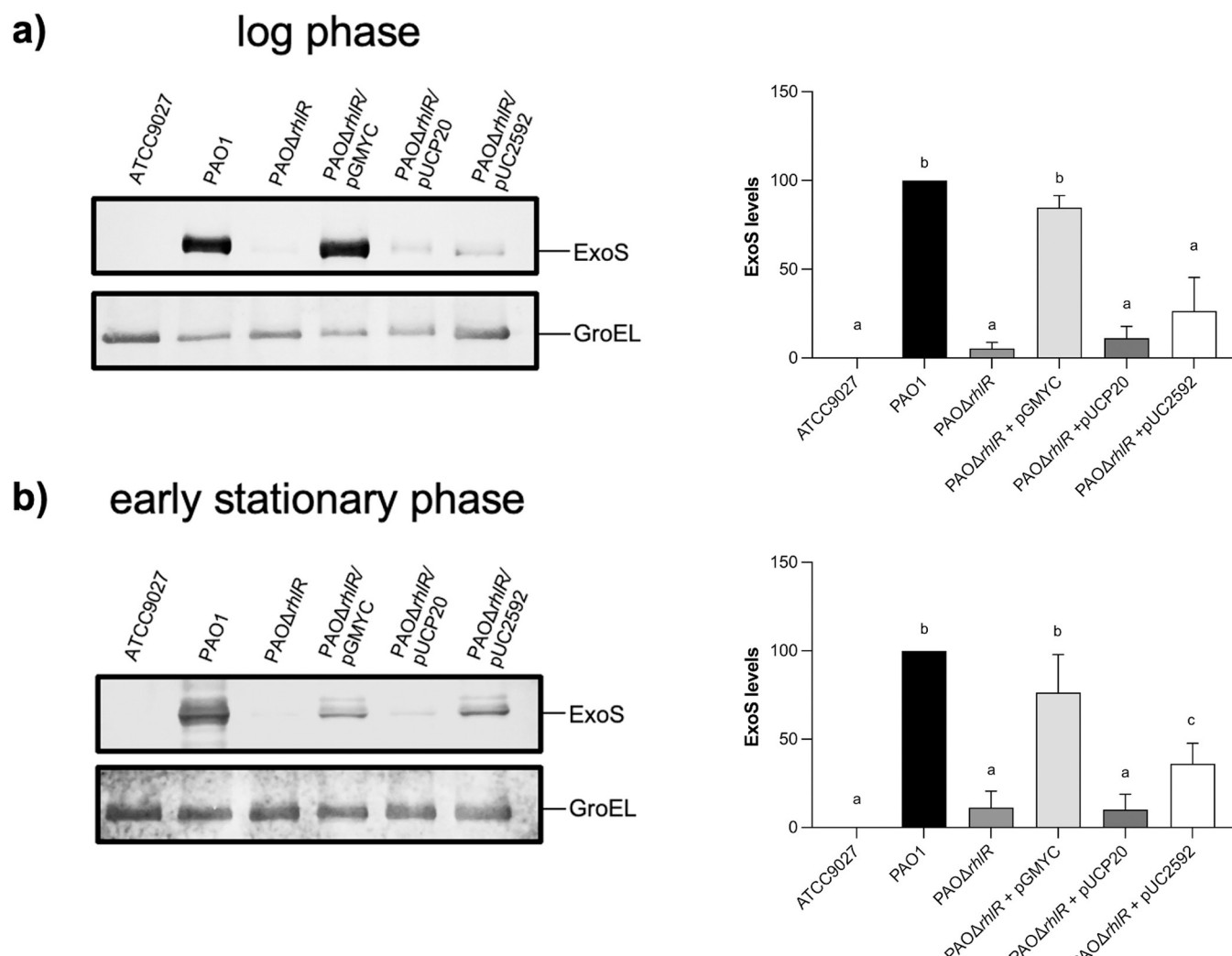

**Fig 7. Effect of *PA2592* expression in the *rhlR* mutant strain.** ExoS identification was performed by Western blot assay using anti-ExoS polyclonal antibody on supernatants of strains grown in induction conditions at log phase (a) and early stationary phase (b). PAO1 was used as a positive control whereas ATCC 9027, lacking the T3SS, was used as a negative control. GroEL was detected using polyclonal antibody anti-GroEL and used as a loading control. The densitometry graphs show the mean ± S.D. of ExoS levels of at least three biological replicates. Significant differences were obtained by ordinary one-way ANOVA and Tukey's multiple comparisons analysis (α = 0.05%). Different letters indicate significant differences, while equal letters indicate no significant differences.

involved in regulating the T3SS since ExoS protein levels in the PAOΔ*lasR* mutant are similar to the ones in the wild-type strain. These results are in agreement with previous reports where expression of T3SS genes is not affected by *lasI* or *lasR* deletion [38,39]. Regarding the Rhl system, we found that the inactivation of *rhlR* almost abolishes ExoS secretion. Moreover, *exoS*, *spcS*, and *exsCEBA* expression are reduced when *rhlR* is inactivated, confirming that RhlR is able to regulate not only *exoS* expression but also other elements of the T3SS, including the master regulator ExsA. Since these results are in contrast to the negative role of RhlR previously proposed [38], we confirmed the positive role of RhlR using another *rhlR* mutant strain, obtaining similar results. This discrepancy on the role of RhlR could be related to the different culture conditions used to activate the T3SS. Moreover, cytotoxicity was affected when *rhlR* was inactivated which supports the positive regulation of RhlR on the T3SS.

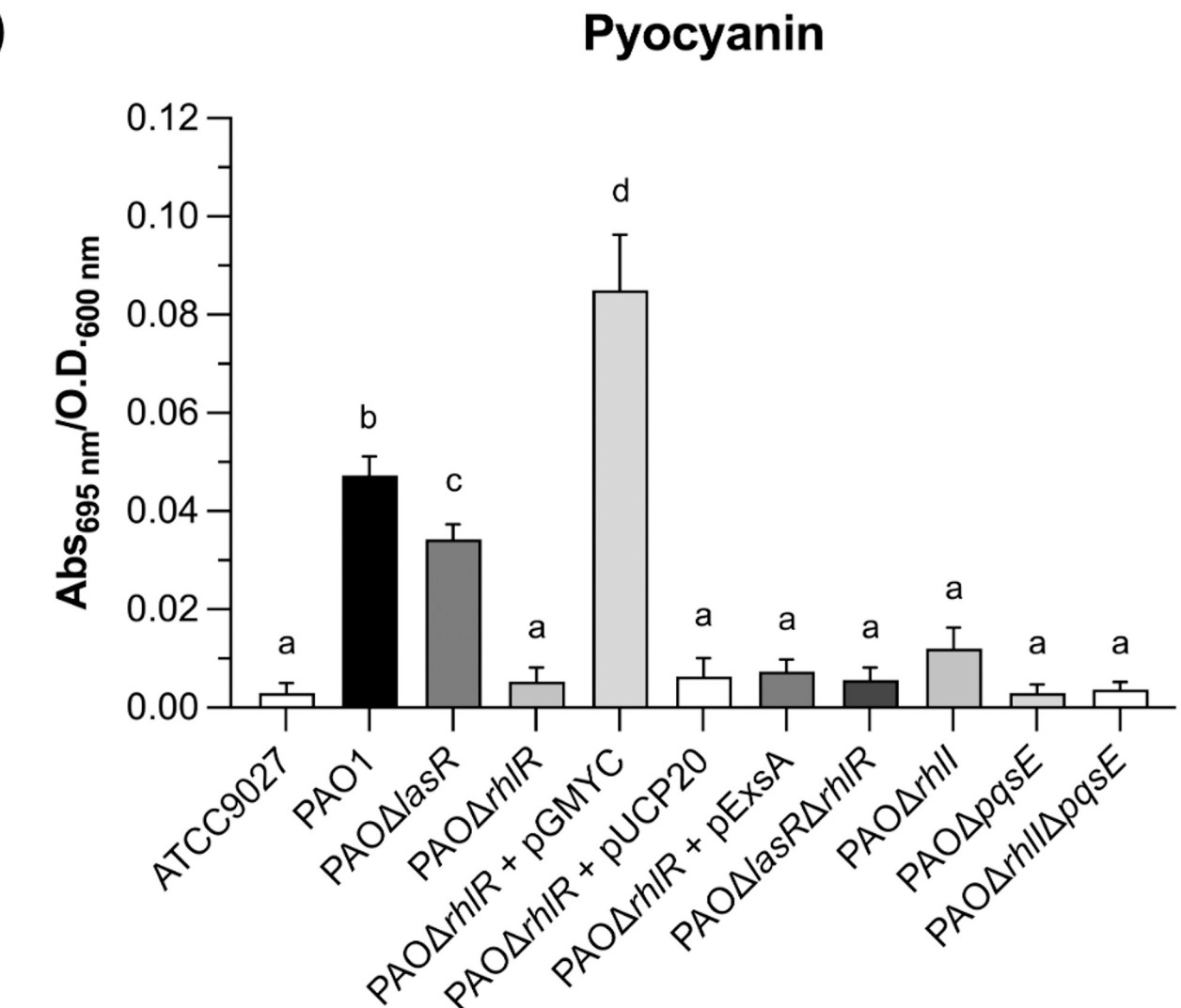

**Fig 8. Pyocyanin production by PAO1 and its derivatives mutant strains in T3SS-induction conditions.** Results represent the mean ± S.D. of three biological experiments performed in duplicates each time. Significant differences were obtained by ordinary one-way ANOVA and Tukey's multiple comparison analysis ($\alpha$ = 0.05%). Different letters indicate significant differences, while equal letters indicate no significant differences.

Additionally, in this study, we used a PAO$\Delta$*lasR*$\Delta$*rhlR* double mutant strain and found that ExoS protein secretion levels are diminished compared to the wild-type strain. This phenotype is explained by the *rhlR* deletion, and this was confirmed when the mutant was complemented with *rhlR* and ExoS secretion was fully restored, resembling a *lasR* mutant strain. In our previous work, we reported that the ExoS secretion was not affected when *lasR* and *rhlR* were inactivated [40]. The mutant strain used in that previous work contained the markers tetracycline and streptomycin (PAO$\Delta$*lasR*$^{Tc}$$\Delta$*rhlR*$^{Sm}$), while the double mutant strain constructed in this work contains the markers apramycin and streptomycin; however, these changes do not explain the differences in the Western blot results. Therefore, we sequenced both genomes in order to find any mutation that could explain these discrepancies. We found that the PAO$\Delta$*lasR*$^{Tc}$$\Delta$*rhlR*$^{Sm}$ strain possesses a single point mutation in the *mvaT* gene that lies in the DNA-binding motif [61] and changes the TGG codon for Trp$^{119}$ to a TAG stop codon. This result was confirmed by sequencing the *mvaT* gene (S8 Fig). MvaT is a negative regulator for the

T3SS that binds to the *exsA* internal promoter and also to the *exsCEBA* operon promoter [19,21], and it has been reported that *mvaT* inactivation derepresses *exsA* expression [21], explaining why in the previously reported PAOΔ*lasR^{Tc}*Δ*rhlR^{Sm}* strain ExoS is detected [40].

RhlR binds to its canonical AI C4 to control the expression of its target genes, however, in recent years it has been reported that RhlR is also able to regulate a set of genes independently of C4 and that this regulation is modulated by PqsE [32,34–36]. With regard to the T3SS, it was reported that *rhlI* inactivation up-regulates the expression of some T3SS genes, including *exoS* and *exoT* expression and ExoS is secreted earlier compared to the wild-type strain [38,39]. Here we showed that, as previously reported, *rhlI* inactivation up-regulates the expression of *exoS* and *exoT* and also the *exsCEBA* operon, indicating that RhlI is a negative regulator of the T3SS. However, our results show that even though the expression of T3SS is increased in the *rhlI* mutant strain, secretion of ExoS is similar to that of the wild-type strain suggesting an additional level of regulation for its secretion. On the other hand, ExoS protein levels in the *pqsE* mutant strain were similar to those of the wild-type strain, and *exsCEBA* transcription was not affected by *pqsE* inactivation, suggesting that PqsE has no role in regulating the T3SS. In addition, ExoS secreted protein levels in the PAOΔ*rhlI*Δ*pqsE* double mutant strain were similar to those of the wild-type strain at 0.8 and 2.0 O.D._{600}, indicating that RhlR is able to activate the T3SS in the absence of RhlI and PqsE. In this regard, it has been reported that RhlR is able to bind to the *rhlAB* promoter and repress its transcription in the absence of C4 [37], demonstrating that RhlR could act as a positive or negative regulator in the presence or the absence of C4 or PqsE. These results also explain why the virulence is reduced when *rhlR* is inactivated but maintained in a *rhlI* mutant strain [32,42].

Since the *exsCEBA* promoter region lacks a site for RhlR, which suggests that this regulation must be in an indirect way, we search for a *las-rhl* box in the previously reported T3SS regulators such as PsrA/RpoS, ArtR, or RetS (also known as RtsM) [5]. However, we could not identify a probable *las-rhl* box in the promoter regions of these regulators. In addition, it has been reported that *rpoS* expression is not regulated by RhlR [62]. Moreover, previous work showed that spermidine is a signal that modulates *exsCEBA* operon expression and, in this context, *PA2592* was identified as a probable gene regulated by RhlR that codes for a putative spermidine-binding protein [22,23,55]. Here, we demonstrated that constitutive expression of *PA2592* in the PAOΔ*rhlR* mutant strain partially restored ExoS secretion only in the early stationary phase. This result indicates that PA2592 has a role in modulating intracellular spermidine concentrations and therefore in regulating the *exsCEBA* operon expression. However, since its expression was unable to fully restore ExoS secretion in the log phase or early stationary phase, it suggests that RhlR controls other elements involved in regulating *exsA* expression.

Finally, since RhlR can control the T3SS expression in the absence of C4 and PqsE, we determined whether C4 and PqsE were still required for pyocyanin production in T3SS-induction conditions as been reported in LB [40,56]. Our results showed that *lasR* inactivation slightly reduced pyocyanin production, but it was abolished in the *rhlR*, *rhlI*, and *pqsE* mutant strains indicating that regulation by the QS systems on pyocyanin synthesis is maintained in T3SS-induction conditions.

Overall, this work permits placing RhlR as a dual regulator, which negatively or positively controls T3SS expression depending upon binding to its C4 autoinducer. This can be related to turning on the T3SS that is necessary at the onset of an acute infection when C4 levels are low but turning it off when C4 levels increase, which also allows to activate the production of other virulence factors dependent on C4 and PqsE such as pyocyanin (Fig 9). These characteristics makes RhlR a very versatile protein that regulates different virulence traits promoting bacterial niche colonization.

**9)**

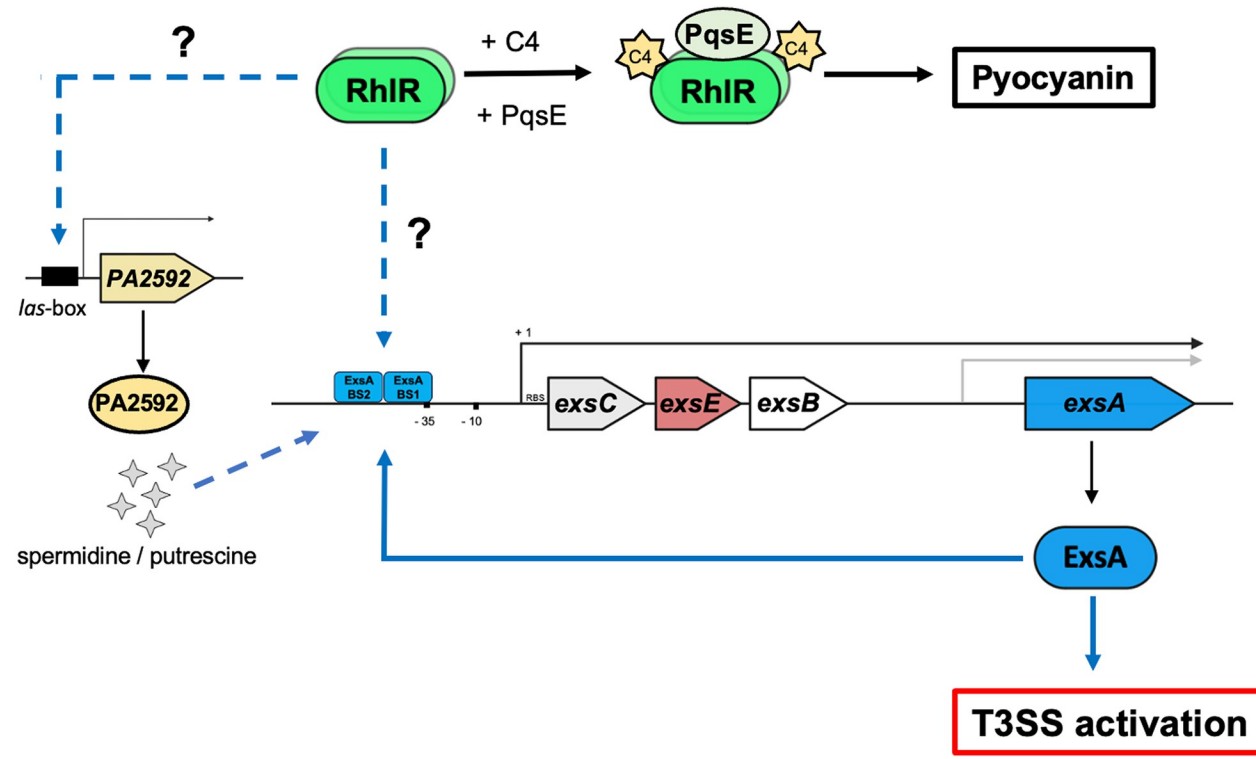

**Fig 9. Regulatory model by RhlR on T3SS-induction conditions.** RhlR with C4 and PqsE positively controls pyocyanin production. In addition, RhlR is a positive regulator of the T3SS; however, this positive regulation is exerted in the absence of C4 and PqsE.

## Supporting information

**S1 Fig. T3SS regulation.** ExsA is the main activator of the T3SS genes. Its activity is controlled by a partner-switching mechanism. During non-induction conditions, ExsD binds to ExsA preventing T3SS activation. Inducing conditions lead to ExsE secretion allowing ExsC to bind ExsD and releasing ExsA, which in turn activates the T3SS expression. Furthermore, *exsA* expression is controlled by additional transcriptional regulators including PsrA, Vfr, MvaT, VqsM, and the post-transcriptional RsmA regulator.
(TIFF)

**S2 Fig. DNA regions used to construct the transcriptional fusions with the *lux* reporter.** DNA regions include -35 and -10 sequences, ExsA binding sites (BS) and/or sites for additional transcriptional regulators previously reported. Nucleotides in base pair (bp) are indicated according to the transcriptional start site (+1).
(TIFF)

**S3 Fig. Effect of *rhlR* expression in the PAOΔ*lasR*Δ*rhlR* double mutant strain on ExoS secretion by Western blot assay at log phase.** ExoS identification was performed by Western blog assay using anti-ExoS polyclonal antibody on supernatants of strains grown in induction conditions. GroEL, detected using polyclonal antibody anti-GroEL, was used as a loading control.
(TIFF)

**S4 Fig. Activation of transcriptional fusions in induction conditions.** Transcriptional fusions activity in PAO1 strain were evaluated in non-induction (LB) and induction conditions (LB + 5 mM EGTA, 20 mM MgCl2). 200 μL of strains were incubated, in triplicate, at an initial O.D.$_{600}$ of 0.05 directly into the wells of a clear-bottomed polystyrene plate, which was incubated at 37˚C without shaking until log phase. Negative control (pCTX), *exsCEBA* operon (PexsC::lux), *exsA* (PexsA::lux), *exoS* (PexoS::lux), and *spsC* (PspcS::lux). Relative luminescence units (R. L.U.) were quantified and normalized to the O.D.$_{600}$ at the time of data collection. Results represent the mean ± S.D. of three biological experiments performed in triplicate each time. Significant differences were obtained by two-way ANOVA and Šídák's multiple comparisons analysis. Asterisks indicate statistical significance (n.s. = not significant, * $p < 0.05$; ** $p < 0.01$; *** $p < 0.001$).
(TIFF)

**S5 Fig. RhlR is unable to activate *exsCEBA* and *exoS* transcription in the absence of ExsA.** The transcriptional activity of *exsCEBA* operon (PexsC::lux) and *exoS* (PexoS::lux) was evaluated in the PAOΔT3SS strain and its derivates with pGMYC or pUCP20 plasmid and in the wild-type PAO1 strain. Strains were incubated in 15 ml of induction medium at 37˚C and 225 rpm until reaching an O.D.$_{600}$ of 0.8 (a) and 2.0 (b). Relative luminescence units (R. L.U.) were quantified and normalized to the O.D.$_{600}$ at the time of cell collection. Results represent the mean ± S.D. of three biological experiments performed in three replicates each time. Significant differences were obtained by two-way ANOVA and Tukey's multiple comparison analysis (α = 0.05%). Different letters indicate significant differences, while equal letters indicate no significant differences.
(TIFF)

**S6 Fig. Effect of *rhlI* and *pqsE* inactivation on *exsCEBA* transcription (PexsC::lux).** Strains were incubated in 15 ml of induction medium at 37˚C and 225 rpm. Relative luminescence units (R. L.U.) were quantified and normalized to the O.D.$_{600}$ at the time of cell collection. Results represent the mean ± S.D. of three biological experiments performed in three replicates each time. Significant differences were obtained by ordinary one-way ANOVA and Tukey's multiple comparison analysis (α = 0.05%). Different letters indicate significant differences, while equal letters indicate no significant differences.
(TIFF)

**S7 Fig. T3SS is activated only during induction conditions.** ExoS identification was performed by Western blot assay using anti-ExoS polyclonal antibody on supernatants of strains grown in non-induction (a) and induction conditions (b). PAO1 was used as a positive control whereas ATCC 9027, lacking the T3SS, was used as a negative control. GroEL was detected using polyclonal antibody anti-GroEL and used as a loading control.
(TIFF)

**S8 Fig. Identification of *mvaT* point mutation G355A in PAOΔ*lasR*$^{Tc}$Δ*rhlR*$^{Sm}$ strain.** Comparison of the alignment of nucleotide (a) and amino acid (b) sequences of strains PAO1 vs PAOΔ*lasR*$^{Tc}$Δ*rhlR*$^{Sm}$ showing a point mutation G355A that generates a stop codon at position Trp119.
(TIFF)

**S1 Table. Strains and plasmids used in this study.**
(DOCX)

**S2 Table. Oligonucleotides used in this study.**
(DOCX)

**S1 Dataset.**
(PDF)

**S1 Raw images.**
(PDF)

## Acknowledgments

LFM-M is a doctoral student of Programa de Maestría y Doctorado en Ciencias Bioquímicas, Universidad Nacional Autónoma de Míxico (UNAM), and thanks CONAHCYT. VRF-V is a master student of Biomedicina Molecular, Centro de Investigación y de Estudios Avanzados del IPN (CINVESTAV-IPN), and thanks CONAHCYT. We thank Dr. Annia Rodríguez Hernández from the Cellular and Molecular Pharmacology Department of the University of California San Francisco (UCSF), for critical reading of the manuscript, and Abigail González Valdez and Norma Espinosa for technical support.

## Author Contributions

**Conceptualization:** Gloria Soberón-Chávez, Miguel Cocotl-Yañez.

**Formal analysis:** Luis Fernando Montelongo-Martínez, Miguel Díaz-Guerrero, Sara Elizabeth Quiroz-Morales, Bertha González-Pedrajo, Gloria Soberón-Chávez, Miguel Cocotl-Yañez.

**Funding acquisition:** Bertha González-Pedrajo, Gloria Soberón-Chávez, Miguel Cocotl-Yañez.

**Investigation:** Luis Fernando Montelongo-Martínez, Miguel Cocotl-Yañez.

**Methodology:** Luis Fernando Montelongo-Martínez, Miguel Díaz-Guerrero, Verónica Roxana Flores-Vega, Martín Paolo Soto-Aceves, Roberto Rosales-Reyes, Sara Elizabeth Quiroz-Morales, Bertha González-Pedrajo.

**Project administration:** Miguel Cocotl-Yañez.

**Supervision:** Miguel Cocotl-Yañez.

**Validation:** Luis Fernando Montelongo-Martínez, Miguel Díaz-Guerrero, Verónica Roxana Flores-Vega, Martín Paolo Soto-Aceves, Roberto Rosales-Reyes, Gloria Soberón-Chávez.

**Writing – original draft:** Luis Fernando Montelongo-Martínez, Miguel Cocotl-Yañez.

**Writing – review & editing:** Luis Fernando Montelongo-Martínez, Verónica Roxana Flores-Vega, Roberto Rosales-Reyes, Bertha González-Pedrajo, Gloria Soberón-Chávez, Miguel Cocotl-Yañez.

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
