## [Decision Letter · Decision Letter 0]

20 Feb 2024

PONE-D-24-00964The quorum sensing regulator RhlR positively controls the expression of the type III secretion system in Pseudomonas aeruginosa PAO1.PLOS ONE

Dear Dr. Cocotl-Yanez,

Thank you for submitting your manuscript to PLOS ONE. After careful consideration, we feel that it has merit but does not fully meet PLOS ONE’s publication criteria as it currently stands. Therefore, we invite you to submit a revised version of the manuscript that addresses the points raised during the review process.

We look forward to receiving your revised manuscript.

Kind regards,

Rajesh P. Shastry, Ph.D

Academic Editor

PLOS ONE

“MC-Y research was supported by Consejo Nacional de Humanidades, Ciencias y Tecnologías (CONAHCYT) FORDECYT‐ PRONACES grant 53366 and Programa de Apoyo a Proyectos de Investigación e Innovación Tecnológica (PAPIIT) DGAPA, Universidad Nacional Autónoma de México (UNAM), grant number IA204221 and IA200823. GS-Ch and BG-P research were supported by DGAPA, PAPIIT UNAM grant IN201222 and IN229023, respectively.”

“LFM‐M is a doctoral student of Programa de Maestría y Doctorado en Ciencias Bioquímicas, Universidad Nacional Autónoma de México (UNAM) and received a fellowship from CONAHCYT (CVU 927093). VRF-V is a master student of Biomedicina Molecular, Centro de Investigación y de Estudios Avanzados del IPN (CINVESTAV-IPN) and received a fellowship from CONAHCYT (CVU No. 1165608). MC-Y research was supported by Consejo Nacional de Ciencia y Tecnología (CONACYT) FORDECYT‐ PRONACES grant 53366 and Programa de Apoyo a Proyectos de Investigación e Innovación Tecnológica (PAPIIT) DGAPA, Universidad Nacional Autónoma de México (UNAM), grant number IA204221 and IA200823. GS-Ch and BG-P research were supported by DGAPA, PAPIIT UNAM grant IN201222 and IN229023, respectively. We thank Dr. Annia Rodríguez Hernández from the Cellular and Molecular Pharmacology Department of the University of California San Francisco (UCSF), for critical reading of the manuscript, and Abigail González Valdez and Norma Espinosa for technical support.”

“MC-Y research was supported by Consejo Nacional de Humanidades, Ciencias y Tecnologías (CONAHCYT) FORDECYT‐ PRONACES grant 53366 and Programa de Apoyo a Proyectos de Investigación e Innovación Tecnológica (PAPIIT) DGAPA, Universidad Nacional Autónoma de México (UNAM), grant number IA204221 and IA200823. GS-Ch and BG-P research were supported by DGAPA, PAPIIT UNAM grant IN201222 and IN229023, respectively.”

Reviewers' comments:

Reviewer's Responses to Questions

**Comments to the Author**

1. Is the manuscript technically sound, and do the data support the conclusions?

Reviewer #1: Yes

Reviewer #2: Yes

2. Has the statistical analysis been performed appropriately and rigorously? 

Reviewer #1: No

Reviewer #2: Yes

3. Have the authors made all data underlying the findings in their manuscript fully available?

Reviewer #1: Yes

Reviewer #2: Yes

4. Is the manuscript presented in an intelligible fashion and written in standard English?

Reviewer #1: Yes

Reviewer #2: Yes

5. Review Comments to the Author

Reviewer #1: The current study investigates the QS essential proteins namely Las, Rhl, and Pqs QS receptors, and inducers' role in the T3SS activation in P. aureginosa. I find these researches that correlate the QS activation to other bacterial activity is an essential area and unfortunately, there are no sufficient studies. However, I have a lot of concerns regarding this study as it requires further deep investigations, I still recommend its publication as it could shed the light in this direction.

1- What is the 1ry antibody used in the western blot

2- I think it would be nice if you can quantify the expresed proteins (use flourescent 2ry antibody and quantify the flourecence, for instance) to obtain statistical significance

3- Line 52, please re-write "Mutant strains defective in this system are severely attenuated in their virulence" with clarification

4- Please clarify the methodology section to be more clear to non-experts, as clarifying the importance of Groel protein as reference one

5- The discussion is well understood, however I recommend writing more detailed conclusion

6- I may recommend the authors to draw a represenative graph illustrating their findings

7- Furthermore, I may advise the authors to extend their findings showing the impact on the bacterial virulence in vitro by quantification of pyocyanin as an example to QS or immunstining to the exoS proteins in macrophages or HeLa cells, as an example

Reviewer #2: This paper examines the role of the global regulator RhlR in the regulation of Pseudomonas aeruginosa type 3 secretion system (T3SS). The transcriptional regulator RhlR was initially characterised at the end of the last century as the receptor for the quorum-sensing signal molecule: N-butanoyl-L-homoserine lactone (C4). However, it is now known that this protein has multiple mechanisms of action, both C4-dependent and C4-independent. The authors re-examine what the role of QS might be in the regulation of T3SS in light of today's knowledge.

Briefly, by means of transcriptional fusions and western-blot experiments, evidence is brought to support the hypothesis that RhlR positively regulates the expression of ExoS (the main effector of T3SS) in a C4-independent manner. Preliminary evidence is also brought to support the hypothesis that RhlR can regulate exoS transcription by positively controlling the promoter of the exsCEBA operon, which in turn encodes for the complex regulatory system that controls T3SS gene expression.

Overall, the authors show that: i) RhlR positively regulates the expression of exoS regardless of the presence of the C4 inducer; ii) this mechanism involves activation of the exsCEBA operon promoter; iii) constitutive transcription in trans of exsA, the main activator of T3SS, restores the expression of exoS in a rhlR mutant.

This is interesting work because it disproves the common belief that QS negatively regulates T3SS expression and draws attention to the multiple mechanisms by which RhlR can control P. aeruginosa virulence. Unfortunately, the work remains somewhat superficial and does not go deeper into the mechanism underlying the RhlR-dependent regulation of exoS and exCEBA genes.

Overall, the work is a bit preliminary and should be enriched with further experiments. Below there are some major issues that the authors need to address in order to get the work published:

1) The regulation of T3SS is very complex, the authors should explain it a little better in the introduction, possibly by including a figure in supplementary showing the exsCEBA operon and the exoS and spcS genes, also indicating the promoters cloned into transcriptional fusions with the lux genes. Concerning T3SS, it would be useful to cite Horna & Ruiz's review (doi:10.1016/j.micres.2021.126719).

2) In Figures 1, 2 and 3, it is not clear whether the experiments were conducted under inducing or not-inducing conditions. In any case, the authors should conduct the experiments shown in Figures 1, 2 and 3 under both conditions.

3) The names given to transcriptional fusions (pS, pP, pC, pA) are too contracted and do not help understanding. Authors should name the promoters as per convention (es. promoter of exoS gene = PexoS. Fusion between PexoS and luxABCDE = PexoS::luxABCDE, the latter could be abbreviated as PexoS::lux).

4) The PexsCEBA promoter is activated by RhlR in a C4- and PqsE-independent manner. Thus, this regulation appears to be indirect. The authors speculate that the regulation might depend on the spermidine binding protein PA2592 (lines 457-463). The authors should construct this mutant and test this hypothesis.

5) The PexsCEBA promoter is regulated by other regulators such as PrsA/RpoS, ArtR, RtsM (Horna & Ruiz of 2021 and references therein). Is there a relationship between these regulators and RhlR? The authors should answer this question referring to literature data or by conducting experiments.

6. PLOS authors have the option to publish the peer review history of their article (what does this mean?). If published, this will include your full peer review and any attached files.

Reviewer #1: **Yes: **Wael A. H. Hegazy

Reviewer #2: No

---

## [Author Response · Author response to Decision Letter 0]

4 Apr 2024

Reviewers' comments:

Reviewer's Responses to Questions

Comments to the Author

1. Is the manuscript technically sound, and do the data support the conclusions?

Reviewer #1: Yes

Reviewer #2: Yes

2. Has the statistical analysis been performed appropriately and rigorously?

Reviewer #1: No. 

R: We carried out an additional statistical analysis to be more rigorous with our results, this analysis has been included in the revised version. 

Reviewer #2: Yes

3. Have the authors made all data underlying the findings in their manuscript fully available?

Reviewer #1: Yes

Reviewer #2: Yes

4. Is the manuscript presented in an intelligible fashion and written in standard English?

Reviewer #1: Yes

Reviewer #2: Yes

5. Review Comments to the Author

Reviewer #1: The current study investigates the QS essential proteins namely Las, Rhl, and Pqs QS receptors, and inducers' role in the T3SS activation in P. aureginosa. I find these researches that correlate the QS activation to other bacterial activity is an essential area and unfortunately, there are no sufficient studies. However, I have a lot of concerns regarding this study as it requires further deep investigations, I still recommend its publication as it could shed the light in this direction.

1- What is the 1ry antibody used in the western blot

R: The first antibody used was anti-ExoS, the information was included in Materials and methods and in the figure legends. 

2- I think it would be nice if you can quantify the expresed proteins (use flourescent 2ry antibody and quantify the flourecence, for instance) to obtain statistical significance

R: Thank you for your suggestion. We carried out a densitometry analysis to obtain statistical significance using at least three Western blot assays and it was included in the Western blot figures. Also, the methodology was included (lines 264-268).

3- Line 52, please re-write "Mutant strains defective in this system are severely attenuated in their virulence" with clarification

R: Thanks for your suggestion. The sentence was modified for clarification (lines 51-52)

4- Please clarify the methodology section to be more clear to non-experts, as clarifying the importance of Groel protein as reference one

R: The methodology section was reviewed and modified to be clear. 

5- The discussion is well understood, however I recommend writing more detailed conclusion.

R: We have written a more detailed conclusion according to your suggestion (lines 562-565)

6- I may recommend the authors to draw a represenative graph illustrating their findings

R: Thank you for your recommendation. A figure summarizing our findings has been included (Fig 9).

7- Furthermore, I may advise the authors to extend their findings showing the impact on the bacterial virulence in vitro by quantification of pyocyanin as an example to QS or immunstining to the exoS proteins in macrophages or HeLa cells, as an example

R: As you suggested, we quantified pyocyanin production in T3SS-induction conditions. This was included in the revised manuscript (lines 258-262 and 458-476).

Reviewer #2: This paper examines the role of the global regulator RhlR in the regulation of Pseudomonas aeruginosa type 3 secretion system (T3SS). The transcriptional regulator RhlR was initially characterised at the end of the last century as the receptor for the quorum-sensing signal molecule: N-butanoyl-L-homoserine lactone (C4). However, it is now known that this protein has multiple mechanisms of action, both C4-dependent and C4-independent. The authors re-examine what the role of QS might be in the regulation of T3SS in light of today's knowledge.

Briefly, by means of transcriptional fusions and western-blot experiments, evidence is brought to support the hypothesis that RhlR positively regulates the expression of ExoS (the main effector of T3SS) in a C4-independent manner. Preliminary evidence is also brought to support the hypothesis that RhlR can regulate exoS transcription by positively controlling the promoter of the exsCEBA operon, which in turn encodes for the complex regulatory system that controls T3SS gene expression.

Overall, the authors show that: i) RhlR positively regulates the expression of exoS regardless of the presence of the C4 inducer; ii) this mechanism involves activation of the exsCEBA operon promoter; iii) constitutive transcription in trans of exsA, the main activator of T3SS, restores the expression of exoS in a rhlR mutant.

This is interesting work because it disproves the common belief that QS negatively regulates T3SS expression and draws attention to the multiple mechanisms by which RhlR can control P. aeruginosa virulence. Unfortunately, the work remains somewhat superficial and does not go deeper into the mechanism underlying the RhlR-dependent regulation of exoS and exCEBA genes.

Overall, the work is a bit preliminary and should be enriched with further experiments. Below there are some major issues that the authors need to address in order to get the work published:

1) The regulation of T3SS is very complex, the authors should explain it a little better in the introduction, possibly by including a figure in supplementary showing the exsCEBA operon and the exoS and spcS genes, also indicating the promoters cloned into transcriptional fusions with the lux genes. Concerning T3SS, it would be useful to cite Horna & Ruiz's review (doi:10.1016/j.micres.2021.126719).

R: Thank you for your suggestion. In this revised version we have included two additional supplementary figures related to the regulation of the T3SS and the transcriptional fusions constructed in this work (Fig S1 and S2). Also, the cite Horna and Ruiz was included. 

2) In Figures 1, 2 and 3, it is not clear whether the experiments were conducted under inducing or not-inducing conditions. In any case, the authors should conduct the experiments shown in Figures 1, 2 and 3 under both conditions.

R: All experiments were conducted under inducing conditions since in LB the T3SS is not active. We have included this information in the revised version and also, we have included a supplementary figure where we showed that ExoS is not detected in non-induction conditions (lines 281-282, 498-500) (Fig S6). 

3) The names given to transcriptional fusions (pS, pP, pC, pA) are too contracted and do not help understanding. Authors should name the promoters as per convention (es. promoter of exoS gene = PexoS. Fusion between PexoS and luxABCDE = PexoS::luxABCDE, the latter could be abbreviated as PexoS::lux).

R: Names of plasmids were changed in the text according to your suggestion. 

4) The PexsCEBA promoter is activated by RhlR in a C4- and PqsE-independent manner. Thus, this regulation appears to be indirect. The authors speculate that the regulation might depend on the spermidine binding protein PA2592 (lines 457-463). The authors should construct this mutant and test this hypothesis.

R: In this work, our main objective was to define the role of the quorum sensing systems on the T3SS expression. Therefore, we did not determine the target of RhlR to control the T3SS expression. However, according to your suggestion, in this revised version, we explored whether PA2595 has a role in the T3SS control. Our approach was to evaluate whether the expression of PA2592 under a constitutive promoter was able to restore ExoS secretion in the rhlR mutant strain. These results were included as a new section (lines 432-456). Also, the discussion was modified according to these results (lines 551-560).

5) The PexsCEBA promoter is regulated by other regulators such as PrsA/RpoS, ArtR, RtsM (Horna & Ruiz of 2021 and references therein). Is there a relationship between these regulators and RhlR? The authors should answer this question referring to literature data or by conducting experiments.

R: Thank you for your comment. There is not a relationship previously reported between RhlR and PsrA, ArtR, or RetS (RtsM). Also, we searched for a las-box in these and other regulators of the T3SS reported by Horna & Ruiz but we could not find any consensus sequence for RhlR. We discussed these findings in the discussion section (lines 547-551).

6. PLOS authors have the option to publish the peer review history of their article (what does this mean?). If published, this will include your full peer review and any attached files.

Do you want your identity to be public for this peer review? For information about this choice, including consent withdrawal, please see our Privacy Policy.

Reviewer #1: Yes: Wael A. H. Hegazy

Reviewer #2: No

---

## [Decision Letter · Decision Letter 1]

14 May 2024

PONE-D-24-00964R1The quorum sensing regulator RhlR positively controls the expression of the type III secretion system in Pseudomonas aeruginosa PAO1.PLOS ONE

Dear Dr. Cocotl-Yanez,

Thank you for submitting your manuscript to PLOS ONE. After careful consideration, we feel that it has merit but does not fully meet PLOS ONE’s publication criteria as it currently stands. Therefore, we invite you to submit a revised version of the manuscript that addresses the points raised during the review process.

We look forward to receiving your revised manuscript.

Kind regards,

Rajesh P. Shastry, Ph.D

Academic Editor

PLOS ONE

Journal Requirements:

Reviewers' comments:

Reviewer's Responses to Questions

**Comments to the Author**

1. If the authors have adequately addressed your comments raised in a previous round of review and you feel that this manuscript is now acceptable for publication, you may indicate that here to bypass the “Comments to the Author” section, enter your conflict of interest statement in the “Confidential to Editor” section, and submit your "Accept" recommendation.

Reviewer #1: All comments have been addressed

Reviewer #3: (No Response)

2. Is the manuscript technically sound, and do the data support the conclusions?

Reviewer #1: Yes

Reviewer #3: Yes

3. Has the statistical analysis been performed appropriately and rigorously? 

Reviewer #1: Yes

Reviewer #3: Yes

4. Have the authors made all data underlying the findings in their manuscript fully available?

Reviewer #1: Yes

Reviewer #3: Yes

5. Is the manuscript presented in an intelligible fashion and written in standard English?

Reviewer #1: Yes

Reviewer #3: Yes

6. Review Comments to the Author

Reviewer #1: (No Response)

Reviewer #3: In this MS Cocotl-Yanez et al. delve into the role of QS in regulating T3SS in Pseudomonas aeruginosa. Since QS has branching global effects on virulence in this pathogen, this is a very relevant topic and the results provided by this group shed some light on T3SS regulation and provide highly needed clarifications to some experimental discrepancies found regarding this topic. The authors make a sound analysis on the regulation of the T3SS and provide solid data that support most of their conclusions. Although I consider it a high quality work and I would support this MS for its acceptance, there are a few concerns that I think need to be addressed regarding some of their conclusions:

Major concerns:

The authors conclude that, as overexpressing exsA restores ExoS production in a rhlR mutant, RhlR controls the T3SS expression via ExsA. This is not necessarily the case, they can both act independently, and overexpressing one of the two independent activators may still correct the lack of the other. Their data proves that RhlR does not act downstream of ExsA, but there is still the question if RhlR acts upstream of ExsA as they claim or if they act in different signaling pathways. This would be clarified if they introduced their rhlR overexpression plasmid in a exsA mutant background. If overproducing RhlR can also restore ExoS production they might be independent activators. On the other hand, if ExoS production is not restored, it may indicate that both regulators act hierarchically as they describe.

I have trouble relating ExoS quantifications in figure 5b with their associated western blot data. What I see in the graphs is not an increase in ExoS protein levels in the rhlI mutant background as claimed in the main text (L399) and glimpsed in the western blot (in any case average levels in the graphs seem to be higher in the pqsE mutant although not significant). Also the statistical analysis does not seem to indicate that any of the mutants besides rhlR is significantly different from the wild type to make any of these statements. Please clarify these discrepancies.

Minor concerns:

Also regarding ExsA and RhlR relationship, it would be interesting to see if the internal PexsA promoter is also affected by the rhlR mutation.

L118: I fail to understand why are there discrepancies on T3SS regulation by the Las system. It is clear why Rhl role in T3SS needs further clarification, but all evidence discussed up to this point about Las system seems to agree that it is not involved. I would suggest changing this to “there are some discrepancies in the virulence and regulation of the T3SS by QS systems” so it is more accurate.

L302: The first section of the results jumps from figure 1 to figure 3. If the authors consider that discussing this mutant at this point of the MS is important I would suggest to rename it as figure 2. Otherwise they can try to introduce this result at a later point.

L395: As they also mentioned in the introduction, RhlR is known to be able to act independently of C4. I would rephrase this to “can be dependent” or other formulation that better reflects that this is not an absolute dependence.

L401-404: It is a bit hard to understand why this result is discussed here, and it also introduces a bit of difficulties in following the reasoning of the experimental procedure. I would move this phrase to a different point, maybe after discussing the results on Fig 6, so it is easier to follow and shows in a clear way why there are differences between exsC promoter activities in Fig 6 and S5.

L438, 442, 551, and 553: The authors probably mean a rhl box or a las-rhl box?

L461: The authors should clarify how their results show dependence on C4 and PqsE. By looking at figure 8 one can see PqsE dependence, but this is not discussed in the text, and C4 dependence is equally not properly described in this section despite being shown in the picture. Please reformulate this whole section so it is clear to the reader why your data supports your claims (which does).

Minor errata and things the authors may want to check before submitting their final version:

L73: control.

L127: show.

L170: remove the primer sequence since it is already in the supplementary table and no other primer sequences are shown.

L265: A citation may be missing here for “as reported previously”.

7. PLOS authors have the option to publish the peer review history of their article (what does this mean?). If published, this will include your full peer review and any attached files.

Reviewer #1: **Yes: **Wael A. H. Hegazy

Reviewer #3: **Yes: **Francisco Javier Marcos-Torres

---

## [Author Response · Author response to Decision Letter 1]

17 Jun 2024

June 17, 2024

Rajesh P. Shastry, Ph.D

Academic Editor

PLOS ONE

In this document, we respond to the comments and suggestions made by the reviewer for the manuscript PONE-D-24-00964R1 entitled ‘The quorum sensing regulator RhlR positively controls the expression of the type III secretion system in Pseudomonas aeruginosa PAO1’ by Luis Fernando Montelongo-Martínez, Miguel Díaz-Guerrero, Verónica Roxana Flores-Vega, Martín Paolo Soto-Aceves, Roberto Rosales-Reyes, Sara Elizabeth Quiroz-Morales, Bertha González-Pedrajo, Gloria Soberón-Chávez, Miguel Cocotl-Yañez. 

Best regards

Miguel Cocotl-Yañez

Corresponding author.

Reviewers' comments:

Reviewer's Responses to Questions

Comments to the Author

1. If the authors have adequately addressed your comments raised in a previous round of review and you feel that this manuscript is now acceptable for publication, you may indicate that here to bypass the “Comments to the Author” section, enter your conflict of interest statement in the “Confidential to Editor” section, and submit your "Accept" recommendation.

Reviewer #1: All comments have been addressed

Reviewer #3: (No Response)

2. Is the manuscript technically sound, and do the data support the conclusions?

Reviewer #1: Yes

Reviewer #3: Yes

3. Has the statistical analysis been performed appropriately and rigorously? 

Reviewer #1: Yes

Reviewer #3: Yes

4. Have the authors made all data underlying the findings in their manuscript fully available?

Reviewer #1: Yes

Reviewer #3: Yes

5. Is the manuscript presented in an intelligible fashion and written in standard English?

Reviewer #1: Yes

Reviewer #3: Yes

6. Review Comments to the Author

Reviewer #1: (No Response)

Reviewer #3: In this MS Cocotl-Yanez et al. delve into the role of QS in regulating T3SS in Pseudomonas aeruginosa. Since QS has branching global effects on virulence in this pathogen, this is a very relevant topic and the results provided by this group shed some light on T3SS regulation and provide highly needed clarifications to some experimental discrepancies found regarding this topic. The authors make a sound analysis on the regulation of the T3SS and provide solid data that support most of their conclusions. Although I consider it a high quality work and I would support this MS for its acceptance, there are a few concerns that I think need to be addressed regarding some of their conclusions:

Major concerns:

The authors conclude that, as overexpressing exsA restores ExoS production in a rhlR mutant, RhlR controls the T3SS expression via ExsA. This is not necessarily the case, they can both act independently, and overexpressing one of the two independent activators may still correct the lack of the other. Their data proves that RhlR does not act downstream of ExsA, but there is still the question if RhlR acts upstream of ExsA as they claim or if they act in different signaling pathways. This would be clarified if they introduced their rhlR overexpression plasmid in a exsA mutant background. If overproducing RhlR can also restore ExoS production they might be independent activators. On the other hand, if ExoS production is not restored, it may indicate that both regulators act hierarchically as they describe.

R: Thank you for your suggestion. We overexpressed rhlR in a strain defective in the T3SS and transcription of exsCEBA and exoS were measured, the results are described in lines 378-387.

I have trouble relating ExoS quantifications in figure 5b with their associated western blot data. What I see in the graphs is not an increase in ExoS protein levels in the rhlI mutant background as claimed in the main text (L399) and glimpsed in the western blot (in any case average levels in the graphs seem to be higher in the pqsE mutant although not significant). Also the statistical analysis does not seem to indicate that any of the mutants besides rhlR is significantly different from the wild type to make any of these statements. Please clarify these discrepancies.

R: Thank you for your observation. We have corrected it (lines 416-417) and also it was discussed (lines 556-561).

Minor concerns:

Also regarding ExsA and RhlR relationship, it would be interesting to see if the internal PexsA promoter is also affected by the rhlR mutation.

R: Thanks for your comment. We found that exsA internal promoter is barely expressed and is not activated by induction conditions, thus we focused on determining the role of the QS systems on the T3SS genes that were activated in induction conditions.

L118: I fail to understand why are there discrepancies on T3SS regulation by the Las system. It is clear why Rhl role in T3SS needs further clarification, but all evidence discussed up to this point about Las system seems to agree that it is not involved. I would suggest changing this to “there are some discrepancies in the virulence and regulation of the T3SS by QS systems” so it is more accurate.

R: It was changed (Line 118)

L302: The first section of the results jumps from figure 1 to figure 3. If the authors consider that discussing this mutant at this point of the MS is important I would suggest to rename it as figure 2. Otherwise they can try to introduce this result at a later point.

R: Figure 3 was renamed as Figure 2. 

L395: As they also mentioned in the introduction, RhlR is known to be able to act independently of C4. I would rephrase this to “can be dependent” or other formulation that better reflects that this is not an absolute dependence.

R: It was changed (Line 414)

L401-404: It is a bit hard to understand why this result is discussed here, and it also introduces a bit of difficulties in following the reasoning of the experimental procedure. I would move this phrase to a different point, maybe after discussing the results on Fig 6, so it is easier to follow and shows in a clear way why there are differences between exsC promoter activities in Fig 6 and S5.

R: The section was modified according to your suggestion. 

L438, 442, 551, and 553: The authors probably mean a rhl box or a las-rhl box?

R: Thank you. It was changed to las-rhl box.

L461: The authors should clarify how their results show dependence on C4 and PqsE. By looking at figure 8 one can see PqsE dependence, but this is not discussed in the text, and C4 dependence is equally not properly described in this section despite being shown in the picture. Please reformulate this whole section so it is clear to the reader why your data supports your claims (which does).

R: Thank you for your comment. The section was modified and also the results were discussed (Lines 586-591).

Minor errata and things the authors may want to check before submitting their final version:

L73: control. R: Corrected

L127: show. R: Corrected

L170: remove the primer sequence since it is already in the supplementary table and no other primer sequences are shown. R: Corrected

L265: A citation may be missing here for “as reported previously”. R: Corrected 

7. PLOS authors have the option to publish the peer review history of their article (what does this mean?). If published, this will include your full peer review and any attached files.

Do you want your identity to be public for this peer review? For information about this choice, including consent withdrawal, please see our Privacy Policy.

Reviewer #1: Yes: Wael A. H. Hegazy

Reviewer #3: Yes: Francisco Javier Marcos-Torres

---

## [Decision Letter · Decision Letter 2]

2 Jul 2024

The quorum sensing regulator RhlR positively controls the expression of the type III secretion system in Pseudomonas aeruginosa PAO1.

PONE-D-24-00964R2

Dear Dr. Cocotl-Yanez,

We’re pleased to inform you that your manuscript has been judged scientifically suitable for publication and will be formally accepted for publication once it meets all outstanding technical requirements.

Kind regards,

Rajesh P. Shastry, Ph.D

Academic Editor

PLOS ONE

Additional Editor Comments (optional):

Reviewers' comments:

Reviewer's Responses to Questions

**Comments to the Author**

Reviewer #1: All comments have been addressed

Reviewer #3: All comments have been addressed

2. Is the manuscript technically sound, and do the data support the conclusions?

Reviewer #1: Yes

Reviewer #3: Yes

3. Has the statistical analysis been performed appropriately and rigorously? 

Reviewer #1: Yes

Reviewer #3: Yes

4. Have the authors made all data underlying the findings in their manuscript fully available?

Reviewer #1: Yes

Reviewer #3: Yes

5. Is the manuscript presented in an intelligible fashion and written in standard English?

Reviewer #1: Yes

Reviewer #3: Yes

6. Review Comments to the Author

Reviewer #1: From the first revision; all the raised points have been adressed and it can be published in the current form

Reviewer #3: With this new version of the MS, Cocotl-Yanez et al have made a remarkable work addressing all previous concerns and strengthening their conclusions. Considering the quality of the resulting work I fully recommend for its publication without further modifications.

7. PLOS authors have the option to publish the peer review history of their article (what does this mean?). If published, this will include your full peer review and any attached files.

Reviewer #1: **Yes: **Wael A. H. Hegazy

Reviewer #3: **Yes: **Francisco Javier Marcos-Torres

---

## [Editor Report · Acceptance letter]

6 Aug 2024

PONE-D-24-00964R2 

PLOS ONE

Dear Dr. Cocotl-Yanez, 

I'm pleased to inform you that your manuscript has been deemed suitable for publication in PLOS ONE. Congratulations! Your manuscript is now being handed over to our production team.

Kind regards, 

on behalf of

Dr. Rajesh P. Shastry 

Academic Editor

PLOS ONE